# TET1 dioxygenase is required for FOXA2-associated chromatin remodeling in pancreatic beta-cell differentiation

Jianfang Li[1,2,3,10], Xinwei Wu[1,8,10], Jie Ke[1,10], Minjung Lee[4], Qingping Lan[1], Jia Li[4,9], Jianxiu Yu[5], Yun Huang[4], De-Qiang Sun[2,6✉] & Ruiyu Xie[1,7✉]

Existing knowledge of the role of epigenetic modifiers in pancreas development has exponentially increased. However, the function of TET dioxygenases in pancreatic endocrine specification remains obscure. We set out to tackle this issue using a human embryonic stem cell (hESC) differentiation system, in which *TET1/TET2/TET3* triple knockout cells display severe defects in pancreatic β-cell specification. The integrative whole-genome analysis identifies unique cell-type-specific hypermethylated regions (hyper-DMRs) displaying reduced chromatin activity and remarkable enrichment of FOXA2, a pioneer transcription factor essential for pancreatic endoderm specification. Intriguingly, TET depletion leads to significant changes in FOXA2 binding at the pancreatic progenitor stage, in which gene loci with decreased FOXA2 binding feature low levels of active chromatin modifications and enriches for bHLH motifs. Transduction of full-length *TET1* but not the TET1-catalytic-domain in *TET*-deficient cells effectively rescues β-cell differentiation accompanied by restoring *PAX4* hypomethylation. Taking these findings together with the defective generation of functional β-cells upon TET1-inactivation, our study unveils an essential role of TET1-dependent demethylation in establishing β-cell identity. Moreover, we discover a physical interaction between TET1 and FOXA2 in endodermal lineage intermediates, which provides a mechanistic clue regarding the complex crosstalk between TET dioxygenases and pioneer transcription factors in epigenetic regulation during pancreas specification.

[1] Department of Biomedical Sciences, Faculty of Health Sciences, University of Macau, Macau SAR 999078, China. [2] Innovation Center for Advanced Interdisciplinary Medicine, the Fifth Affiliated Hospital of Guangzhou Medical University, Guangzhou 510530, China. [3] Guangzhou Laboratory, Guangzhou 510005, China. [4] Center for Epigenetics & Disease Prevention, Institute of Biosciences and Technology, College of Medicine, Texas A&M University, Houston, TX 77030, USA. [5] Department of Biochemistry and Molecular Cell Biology & Shanghai Key Laboratory of Tumor Microenvironment and Inflammation, Shanghai Jiao Tong University School of Medicine, Shanghai 200025, China. [6] Cardiology Department, the Second Affiliated Hospital, Zhejiang University School of Medicine, Hangzhou 310009, China. [7] Ministry of Education Frontiers Science Center for Precision Oncology, University of Macau, Macau SAR 999078, China. [8] Present address: Thoracic Epigenetics Section, Thoracic Surgery Branch, Center for Cancer Research, National Cancer Institute, National Institutes of Health, Bethesda, MD 20892, USA. [9] Present address: State Key Laboratory of Respiratory Disease, National Clinical Research Center for Respiratory Disease, Guangzhou Institute of Respiratory Health, the First Affiliated Hospital of Guangzhou Medical University, Guangzhou 510120, China. [10] These authors contributed equally: Jianfang Li, Xinwei Wu, Jie Ke. ✉email: deqiangs@zju.edu.cn; ruiyuxie@um.edu.mo

During embryonic development, pluripotent human embryonic stem cells (hESCs) differentiate into many diverse lineages that make up the complex tissues and organs of the human body. Pancreatic lineage specification relies on the crosstalk between genome and environmental cues in the progenitor niche. This crosstalk is mediated by cis-regulatory elements that play a prominent role in spatiotemporal gene regulation during embryogenesis. In particular, distal regulatory elements, such as enhancers, serve as information integration hubs that allow binding of multiple regulators, including lineage-specific transcription factors (TFs) as well as epigenetic readers, writers, and erasers to ensure integration of intrinsic and extrinsic environmental cues at these loci[1,2]. It was recently demonstrated that pioneer TFs, such as FOXA1 and FOXA2, are required for proper chromatin opening and establishment of enhancer marks H3K4me1 and H3K27ac during pancreatic fate specification[3,4]. These pioneer TFs bind to nucleosomal DNA to initiate chromatin remodeling associated with DNA demethylation at newly accessible enhancers[5–8].

DNA demethylation is mediated by ten-eleven-translocation methylcytosine dioxygenases (TETs), which catalyze sequential oxidation of 5-methylcytosine (5mC) to 5-hydroxymethylcytosine (5hmC), 5-formylcytosine, and 5-carboxylcytosine[9–13]. Distribution of 5hmC, a novel epigenetic modification, is dynamically changed by and positively correlated with active gene transcription during early lineage specification[14–16]. Consequently, inhibition of TET family enzymes (TET1, TET2, and TET3) impairs cell fate commitment into neural, hematopoietic, cardiac, and several other lineages[17–19]. We previously demonstrated that 5hmC positively correlates with 'open' chromatin at poised and active enhancers in multiple endodermal lineage intermediates using a stepwise hESC differentiation system toward pancreatic progenitors[20]. However, the functional relevance and mechanisms by which TETs regulate pancreatic endocrine specification are currently unclear. While it is recently recognized that pioneer TF FOXA2 plays a critical role in enhancer activation during hESC pancreatic differentiation[3,4,21], there is still scarce understanding of whether and how FOXA2 interacts with TETs to mediate the opening of surrounding chromatin.

Here, we have tackled those questions by studying the biological roles of TETs in the context of in vitro pancreas differentiation from hESCs. We have found that TET1/TET2/TET3 triple knockout cells display severe defects in pancreatic β-cell specification accompanied by reduced chromatin activity and FOXA2 binding at cell-type-specific hypermethylated regions (hyper-DMRs). We further show that TET1 is required for the generation of functional insulin-producing cells, while FOXA2 physically interacts with TET1 and therefore contributes to specific recruitment of TET1 to mediate chromatin opening at the regulatory elements of lineage determinants. Thus, our study unveils complex crosstalk between TET dioxygenases and pioneer TF FOXA2 in establishing β-cell identity.

## Results

### TET deficiency impairs pancreatic endoderm differentiation.
To determine the biological significance of TET-dependent regulation during pancreas specification, we generated TET1, TET2, and TET3 single knockout (KO), double knockout (DKO), and triple knockout (TKO) H1 hESC lines using CRISPR/Cas9 technology (Supplementary Fig. 1a). Mutations resulted in premature termination codons, which were confirmed by Sanger sequencing. Global DNA hydroxymethylation levels from positive clones were assayed by 5hmC dot blot. A significant reduction of 5hmC signals was shown in TKO, TET1KO, TET1/2DKO, and TET1/3DKO cells, whereas TET2KO, TET3KO, and TET2/3DKO cells showed minimal alterations of 5hmC levels (Supplementary

Fig. 1b). To avoid functional redundancy, we focused our initial analysis on TKO cells devoid of any TET-mediated active demethylation. Consistent with other TET-knockout mESC and hESC lines[17,22], TKO H1-hESCs exhibited no apparent defects in stem cell self-renewal capacity or expression of pluripotent factors (Supplementary Fig. 1e).

To determine whether TET proteins affect hESC differentiation toward pancreatic endocrine fate, we used an established stepwise differentiation platform[20] to induce efficient differentiation of hESCs to definitive endoderm (DE), primitive gut tube (GT), pancreatic progenitor (PP), and pancreatic endocrine (PE) (Fig. 1a). Both TET triple-deficient lines (clones 2 and 6) displayed similar differentiation efficiency toward endoderm germ layer as the wild-type (WT) hESC line, in which over 90% of cells were SOX17[+] by day 3 of differentiation (Supplementary Fig. 1c). Moreover, expression levels of other endoderm markers, such as FOXA2, FOXA1, and CXCR4, were unchanged in TET-depleted cells compared with control cells (Supplementary Fig. 1d, e). This analysis suggests that TET dioxygenases are dispensable for endoderm specification in the context of in vitro hESC differentiation. To examine the effects of TET ablation on pancreatic commitment, we subsequently examined the expression of critical pancreatic markers at the PP and PE stages. Using flow cytometry to quantitate the expression of PDX1 at the PP stage and co-expression of PDX1 and NKX6.1 at the PE stage, we observed a substantial decrease in TET-knockout cells (Fig. 1b). These data were confirmed by immunofluorescence staining and RT-qPCR analysis (Fig. 1c, Supplementary Fig. 1e). Consistent with these findings, expression of the pancreatic progenitor markers SOX9 and PTF1A and endocrine hormones insulin, and glucagon were significantly downregulated in TKO lines (Fig. 1c, Supplementary Fig. 1e). Interestingly, PAX4, a key determinant for β-cell specification, failed to induce in TET-inactivated cells, whereas expression of the α-cell determinant, ARX, was not inhibited (Supplementary Fig. 1e). These results demonstrate that TETs are required for pancreatic β-cell lineage specification.

To investigate TET-dependent global transcriptional changes during pancreatic endocrine cell fate commitment, we performed transcriptome analysis of WT and TKO cells at the ES, DE, GT, and PP stages. Principal component analysis of transcriptome data demonstrated that TET-deficient cells were similar at the ES, DE, and GT stages but differed substantially at the PP stage (Fig. 1d). Hierarchical clustering of genes specifically expressed at each stage revealed a remarkable downregulation of PP-specific genes in TKO_PP cells, whereas ES-specific and DE-specific genes showed no obvious changes in TET-deficient cells compared with WT cells (Supplementary Fig. 1f). In particular, 1173 differentially expressed genes (DEGs) were identified in TKO_PP cells (Fig. 1e, Supplementary Data 1). Among them, there were 921 downregulated DEGs in TET-deficient cells, including many important developmental determinants for pancreatic endocrine specification, such as PDX1, NKX2.2, NKX6.1, NKX6.2, NEUROG3, and PAX4. We also observed 252 genes, particularly HOX family members, which were significantly upregulated upon TET deletion, suggesting that failure to differentiate into pancreatic β-cells is tightly linked to an aberrant TF network. In addition, KEGG pathway analysis of the DEGs showed enrichment for terms associated with maturity-onset diabetes of the young (MODY) and insulin secretion (Fig. 1f, Supplementary Data 2), implicating that loss of TETs impairs pancreatic endocrine formation at later stages of endoderm specification.

### FOXA2 binding is enriched at distal regulatory elements displaying decreased accessibility.
Given that our previous study demonstrated that TET-mediated cytosine oxidation is strongly

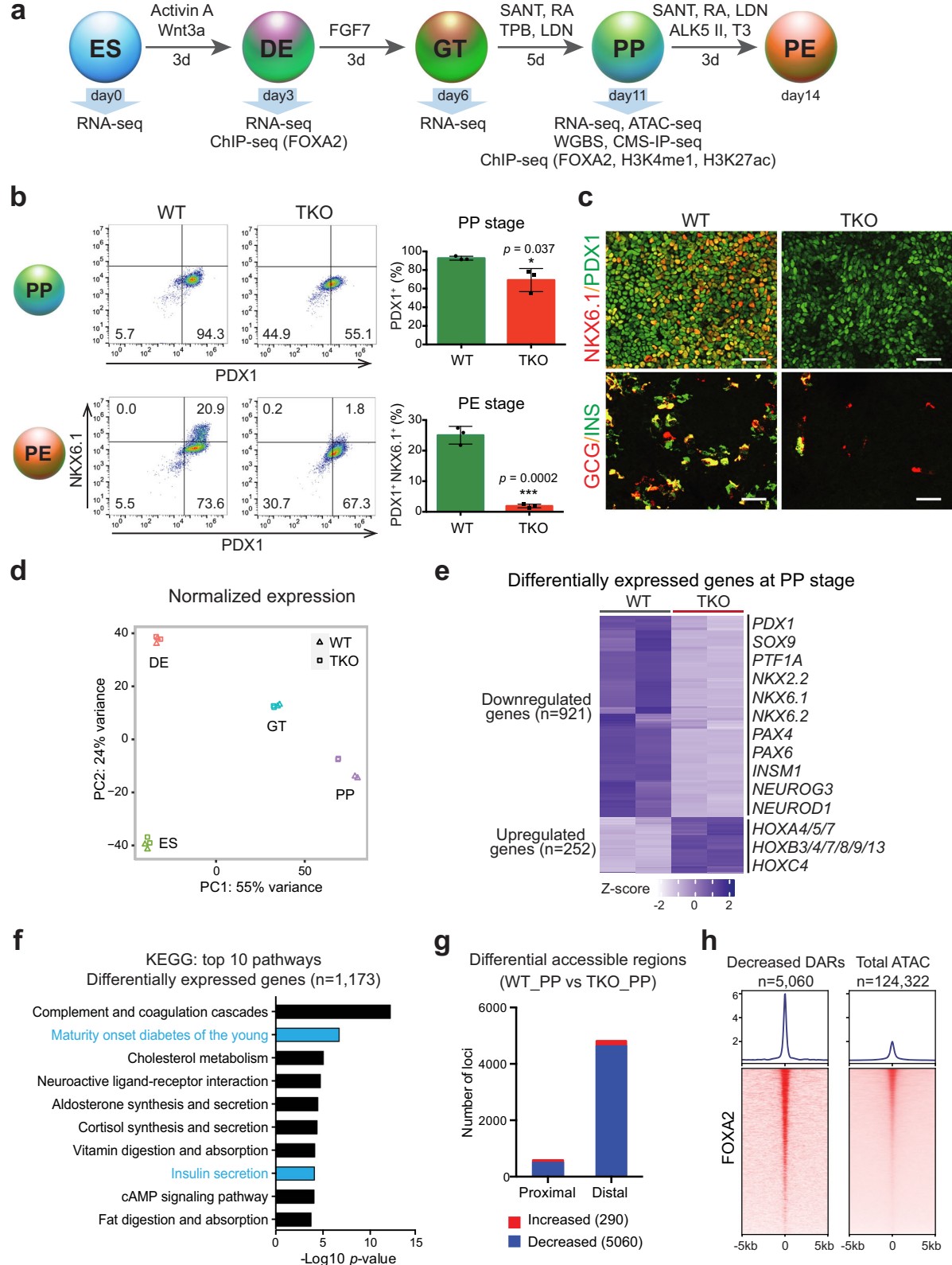

associated with open chromatin regions during pancreatic differentiation[20], we used the assay of transposase-accessible chromatin followed by high-throughput sequencing (ATAC-seq) to evaluate chromatin accessibility landscapes in WT and TET-deficient cells. We assessed differential accessible regions (DARs) between WT_PP and TKO_PP cells from a total of 124,322 identified ATAC-seq peaks and found that 4.3% of ATAC-seq

peaks significantly changed upon TET deletion (Supplementary Data 3). We found that loss of TET led to an overall reduction in chromatin accessibility, with 5060 DARs showing reduced accessibility and only 290 DARs showing increased accessibility in TKO_PP cells (Supplementary Fig. 1g). Moreover, DARs with decreased accessibility were located primarily at distal regions (>1 kb transcription start site (TSS)) (Fig. 1g).

**Fig. 1 TET hydroxylases are required for pancreatic endocrine lineage specification. a** Stepwise differentiation of hESCs to pancreatic endocrine cells. Wild-type (WT) and TET1/TET2/TET3 triple knockout (TKO) hESC lines were differentiated and analyzed by RNA-seq, WGBS, CMS-IP-seq, ATAC-seq, and ChIP-seq at the indicated stages (ES embryonic stem cell, DE definitive endoderm, GT primitive gut tube, PP pancreatic progenitor, PE pancreatic endocrine). **b** Representative flow cytometry plots for PDX1 and NKX6.1 in WT or TKO cells at the PP and PE stages. Quantification of the percentage of PDX1+ NKX6.1+ cells is shown in the right panel (n = 3 independent differentiations; student's t-test, 2-sided; without multiple test correction). All bar graphs show mean ± SD. **c** Immunostaining of PDX1/NKX6.1 at the PP stage and insulin (INS)/glucagon (GCG) at the PE stage are shown in WT and TKO cells (n = 6 independent differentiations; scale bar = 50 μm). **d** Principal component analysis showing variance in normalized transcriptome between WT and TKO cells at the ES, DE, GT, and PP stage. Each plotted point represents one biological replicate. **e** Genes with significant changes in expression in TKO cells relative to WT cells at the PP stage. Differentially expressed genes (DEGs) are classified into down- and upregulated groups. Each row corresponds to one individual gene and each column to a different biological replicate. The color scale from white to blue represents Z-score normalized gene expression levels from low to high (|fold change| ≥ 2; FDR < 0.05). **f** Functional analysis of DEGs in TKO_PP cells showing the top 10 KEGG pathways. Benjamini-Hochberg corrected p-values were used. **g** Number of regions with significant changes in chromatin accessibility upon TET depletion at proximal (≤1 kb from TSS) and distal (>1 kb from TSS) regions (FDR < 0.05). **h** Average density plots (top) and heatmaps (bottom) of FOXA2 binding at decreased accessible regions (DARs) and total identified ATAC-seq peaks (±5 kb). Heatmaps are ranked by decreased FOXA2 binding. The color scale from white to red represents the normalized signal from low to high.

To enhance the biological insights of TET depletion-mediated changes in chromatin accessibility, we analyzed the presence of TF-binding motifs within DARs and found that regions with reduced accessibility were mostly enriched for the motif of pioneer TF FOXA2 essential for pancreas organogenesis and chromatin remodeling[4,21,23,24] (Supplementary Fig. 1h, Supplementary Data 4). Since the presence of a binding motif is not necessarily reflected of actual binding, we mapped FOXA2-binding sites by chromatin immunoprecipitation followed by next-generation sequencing (ChIP-seq) in pancreatic progenitors. Consistent with the motif prediction our analysis revealed that FOXA2 binding was enriched at ATAC-seq peaks preferentially decreased upon TET depletion (Fig. 1h). Collectively, our data indicate that failure to differentiate to pancreatic β-cells is likely to be linked with the loss of chromatin accessibility at distal regulatory elements enriched for pioneer TF FOXA2.

**Differentiation-specific DMRs feature reduced chromatin activity at pioneer TF targets.** To investigate TET deficiency-mediated alterations in global DNA methylation, we performed whole-genome bisulfite sequencing (WGBS) in WT_PP and TKO_PP cells. Over 86% of sequencing reads were uniquely aligned to hg38 with a high sequencing depth of 24 × on CpG dinucleotides and typical bimodal distribution of methylation ratio in each sample (Supplementary Fig. 2a, b). The methylation ratio (5mC/C) was depleted at TSS but enriched across gene coding regions, especially within TKO_PP cells (Supplementary Fig. 2c). Among 26.6 million CpG sites detected in both WT_PP and TKO_PP cells (depth ≥ 5), a total of 251,658 differentially methylated cytosines (DMCs; credible difference of methylation ratio >20%) were identified (Fig. 2a). Strikingly, 97.5% of DMCs were hypermethylated (hyper-DMCs), suggesting that TET inhibition results in a pronounced gain of methylation during pancreas differentiation. We observed enrichment of DMCs primarily at intergenic regions and introns (Supplementary Fig. 2d), whereas substantial enrichment of hypo-DMCs was also found in repeat elements, such as long interspersed elements. Based on an established link between DNA methylation and transcription regulation, we further calculated changes in methylation levels on *cis*-regulatory elements previously identified in pancreatic progenitors[3,25]. We found increased methylation at bivalent promoters, active enhancers, and, to a lesser extent, poised enhancers (Fig. 2b). Notably, hypermethylation was preferentially enriched at open chromatin (ATAC-seq) as well as distal binding regions of pioneer TFs FOXA2, GATA4 (GSE117136)[4], and GATA6 (GSE117136)[4] (Fig. 2b, Supplementary Fig. 2e). This analysis demonstrates that active regulatory elements are hypermethylated upon TET depletion during pancreatic differentiation.

Our previous studies reveal that DNA demethylation is correlated with pancreatic endocrine patterning[20]. To gain better insight into the role of TET-mediated methylation changes in pancreatic differentiation, we identified the differentially methylated regions (DMRs) between TKO_PP and WT_PP cells by connecting at least three consecutive DMCs with a distance between two consecutive DMCs <300 bp[26]. We found a total of 16,490 hyper-DMRs and classified them into two categories based on their 5mC levels in pancreatic progenitor versus hESC[27] (Fig. 2c). Notably, more than half (n = 10,254) of hyper-DMRs exhibited decreased 5mC in WT_PP cells than in hESCs (Supplementary Fig. 2f, Supplementary Data 5) indicating an active demethylation process during pancreatic differentiation (named differentiation-specific hyper-DMRs), whereas the others (n = 6236) showed similar methylation levels (non-differentiation hyper-DMRs) (Fig. 2c, Supplementary Data 5). Interestingly, we found that pioneer TFs FOXA2, GATA4, and GATA6 were substantially enriched at differentiation-specific hyper-DMRs, while less enrichment of pancreatic lineage-specific TFs such as PDX1 (GSE117136)[4] and HNF6 (GSE149148)[28] was observed (Fig. 2d), suggesting that TET-dependent demethylation during pancreatic cell fate commitment occurs at loci primarily associated with pioneer TFs. We further compared ATAC-seq peaks at differentiation-specific hyper-DMRs with those at non-differentiation hyper-DMR. We found that chromatin accessibility was markedly reduced in TET-deficient cells at distally located hyper-DMRs in the differentiation-specific group (Fig. 2e, Supplementary Fig. 2g). Consistently, genomic loci displaying decreased accessibility showed a substantial increase in DNA methylation upon depletion of TET (Fig. 2f). For instance, the hyper-DMRs, which were annotated to PDX1 and its lncRNA PDX1-AS1 loci, overlapped with pioneer TF-binding sites and showed decreased chromatin activity in a TET-dependent manner (Fig. 2g). Significant hypermethylation was also found at the distal regulatory elements of NEUROD1 and NKX2.2 loci (Supplementary Fig. 2h). In summary, these analyses demonstrate that TET-dependent demethylation at TF-enriched distal regulatory elements is essential for chromatin remodeling during pancreas development.

**TET deficiency induces a differentiation-specific loss of 5hmC at FOXA2 target sites.** To get additional insight into the TET-mediated hydroxymethylation network governing pancreatic differentiation, we used cytosine-5-methanesulfonate immunoprecipitation (CMS-IP) coupled with high-throughput sequencing to profile 5hmC landscapes in WT_PP and TKO_PP cells. As a consequence of TET deletion, we found genome-wide loss of hydroxymethylation in TET-deficient cells (Supplementary Fig. 3a,

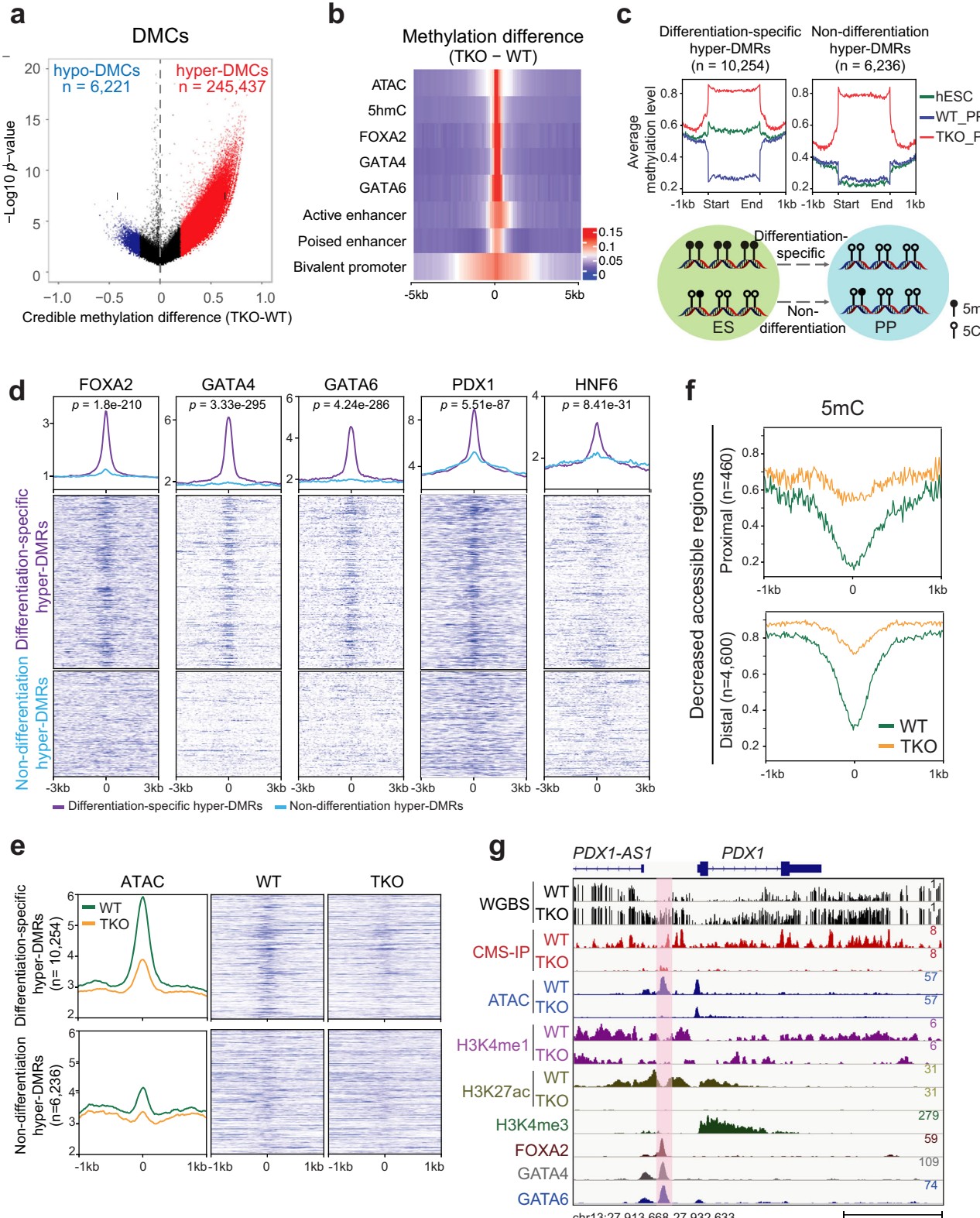

b). As expected, 94% of CMS-positive regions showed a significant reduction in TKO_PP cells relative to WT_PP cells (Supplementary Fig. 3c). These differentially hydroxymethylated regions displaying reduced 5hmC (hypo-DHMRs) were primarily located at non-promoter regions (>1 kb from TSS) (Supplementary Fig. 3d).

Reduced DNA hydroxymethylation in TET-deficient cells could be a result of the initial loss of 5hmC at the ES stage or

unsuccessful oxidation of 5mC during lineage progression. To provide biological insight into differentiation-specific changes of hydroxymethylation upon TET inactivation, we systematically analyzed hypo-DHMRs overlapped with previously identified regions showing de novo increased 5hmC during hESC pancreatic differentiation[20]. Specifically, 27,113 hypo-DHMRs with higher 5hmC levels in WT_PP than in WT_ES (named

**Fig. 2 Differentiation-specific hyper-DMRs show reduced chromatin activity at pioneer TF-binding sites. a** Volcano plot of WGBS data illustrating differentially methylated CpGs (DMCs) identified in TKO_PP compared with WT_PP. Red and blue represent increased and decreased 5mC in TKO_PP cells, respectively (credible methylation difference >0.2). **b** Heatmap illustrating methylation difference between TKO_PP and WT_PP at centers of annotated genomic features (±5 kb) for chromatin accessibility (ATAC), hydroxylation (5hmC), TF binding (FOXA2, GATA4, and GATA6), bivalent promoters, poised enhancers, and active enhancers. Average 5mC signals of every 100-bp bin were calculated. **c** Classification of TKO hyper-DMRs based on 5mC levels in hESCs (green), WT_PP cells (blue), and TKO_PP cells (red). **d** Average density plots and heatmaps of FOXA2, GATA4, GATA6, PDX1, and HNF6 signals at differentiation-specific hyper-DMRs or non-differentiation hyper-DMRs in pancreatic progenitors. The statistical significance was calculated using the student's unpaired two-tailed t-test without multiple test correction. **e** Average density plots and heatmaps of ATAC-seq reads at differentiation-specific hyper-DMRs or non-differentiation hyper-DMRs in PP for WT (green) or TKO (orange) cells. **f** Enrichment profile of methylation ratio (5mC/C) at proximal (≤1 kb from TSS) and distal (>1 kb from TSS) decreased accessible regions in PP for WT (green) and TKO (orange) cells. **g** Genome-browser view of the *PDX1/PDX1-AS1* locus. A specific TKO hyper-DMR showing decreased 5hmC, ATAC-seq, and H3K27ac signals is highlighted in pink.

'differentiation-specific hypo-DHMRs') were first isolated from a total of 70,857 TKO hypo-DHMRs (Fig. 3a). We subsequently clustered them into four categories based on the dynamic changes of 5hmC across stages in WT cells (Fig. 3b, Supplementary Data 6). In particular, one group gained 5hmC specifically at the PP stage (PP-specific), whereas others gained 5hmC at the earlier DE (DE-to-PP) or GT (GT-to-PP) stages and sustained hydroxymethylation until the PP stage. We found that GATA motifs were mostly associated with the DE-to-PP cluster, in which gain of 5hmC began at the DE stage (Fig. 3c, purple).

By contrast, the PP-specific cluster displayed a prominent presence of binding sites for HNF6, which is critical for pancreatic endocrine differentiation[29] (Fig. 3c, red). Most notably, FOXA motifs were predominantly associated with the GT-to-PP(h) cluster in which 5hmC levels increased at the GT stage and continued to be elevated at the PP stage (Fig. 3c, blue), whereas both FOXA and GATA motifs were highly associated with the GT(h)-to-PP cluster in which 5hmC peaked at the GT stage and subsequently decreased at the PP stage (Fig. 3c, brown). Collectively, these results suggest a unique temporal binding pattern of GATAs, FOXAs, and HNF6s associated with the dynamic distribution of 5hmC during pancreatic differentiation, which is supported by a recent study demonstrating sequential requirements for these TFs during transitions in pancreas development[4].

**5hmC and FOXA2 co-occupy sites are essential for chromatin opening**. Given that hydroxymethylation is positively associated with chromatin activity[20] and FOXA2 is strongly enriched in regions featured decreased chromatin accessibility in TET-deficient cells (Fig. 1h), we wondered if binding of FOXA2 dynamically correlated with TET-mediated hydroxymethylation. Thus, we mapped FOXA2-binding sites by ChIP-seq at the DE, GT, and PP stages where FOXA2 continually expressed. Consistent with the motif predictions our analysis revealed that FOXA2 deposition corresponded well with the dynamic changes of 5hmC across differentiation stages in the four 'differentiation-specific hypo-DHMRs' clusters (Fig. 3d). Together, these data support that FOXA2 binding is highly dynamic during pancreatic differentiation and strongly prefers differentiation-associated 5hmC deposition sites.

It was previously demonstrated that the pioneer TF FOXA2 is essential for enhancer priming during pancreatic differentiation[3,4]. We, thus, speculated that loss of TETs inhibits enhancer activation, particularly at hypo-hydroxymethylated regions enriched with FOXA2 binding. We conducted ChIP-seq for enhancer signatures H3K4me1 and H3K27ac in WT_PP and TKO_PP cells and performed integrated analysis. We found a remarkable decrease in ATAC-seq and H3K27ac signal at 'differentiation-specific hypo-DHMRs' overlapped with FOXA2 peaks, where DNA methylation was increased upon TET

depletion (Fig. 3e). Nevertheless, FOXA2 and other pioneer TFs expressed at similar levels in TKO and WT cells (Supplementary Fig. 3e), implicating that inhibition of TET did not alter their expression. Collectively, these data reveal that TET deficiency results in a progressive failure of 5mC oxidation at a subset of FOXA2 targets essential for the establishment of active enhancers during differentiation.

**De novo recruitment of FOXA2 to differentiation stage-specific genomic loci**. Being a pioneer TF, FOXA2 is expected to engage its binding sites on nucleosomal DNA to mediate nucleosome depletion and chromatin remodeling[6,30]. Despite its potentially universal binding ability, FOXA2 targeting was highly dynamic between pancreatic differentiation states (Fig. 3d). We, therefore, ask if additional factors can influence the binding of FOXA2. We tackled these questions by examining FOXA2 occupancy in WT and TKO cells at the DE stage, where TET depletion resulted in minimal phenotypic and transcriptional changes, and at the PP stage in which TET knockout impaired pancreatic cell fate commitment. Consistent with previous reports[4,31], FOXA2 was primarily located at non-promotor regions (Supplementary Fig. 4a) implicating that FOXA2 is mainly involved in transcription regulation through distal regulatory elements. Upon TET depletion, 99.5% of FOXA2 binding sites did not show changes at the DE stage (Fig. 4a, Supplementary Data 7), while FOXA2 occupancy was changed at the PP stage, in which 10% and 6% of FOXA2 target sites showed reduced and greater FOXA2 binding, respectively (Fig. 4a, Supplementary Fig. 4b, Supplementary Data 8). Since DNA methylation and hydroxymethylation status were significantly altered in TET-deficient cells (Fig. 2a, Supplementary Fig. 3c), we wondered whether increases in methylation or decreases in hydroxymethylation contributed to differential recruitment of FOXA2 in TKO_PP cells. We first assessed changes in FOXA2 binding in WT_PP versus TKO_PP at the hyper-DMRs or hypo-DHMRs identified in TKO_PP cells. We found that ~75% of regions showed no changes in FOXA2 binding (Fig. 4b). We then tested if only hypermethylated or hypo-hydroxymethylated FOXA2-bound sites would be affected. We found that similar proportions of FOXA2 target sites displayed reduced/greater FOXA2 binding regardless of the presence or absence of hypermethylation (Fig. 4c). Similarly, hypo-hydroxymethylated FOXA2-bound sites did not show a preference in gain/loss of FOXA2 binding compared to iso-hydroxymethylated FOXA2-bound sites (Fig. 4d). Consistent with the ability of pioneer TFs to engage with inaccessible chromatin, our results demonstrate that DNA methylation/hydroxymethylation states do not interfere in FOXA2 binding.

We next investigated local chromatin landscape at the FOXA2-decreased and -increased sites. Analysis of chromatin accessibility, H3K27ac, and H3K4me1 signal intensity revealed more

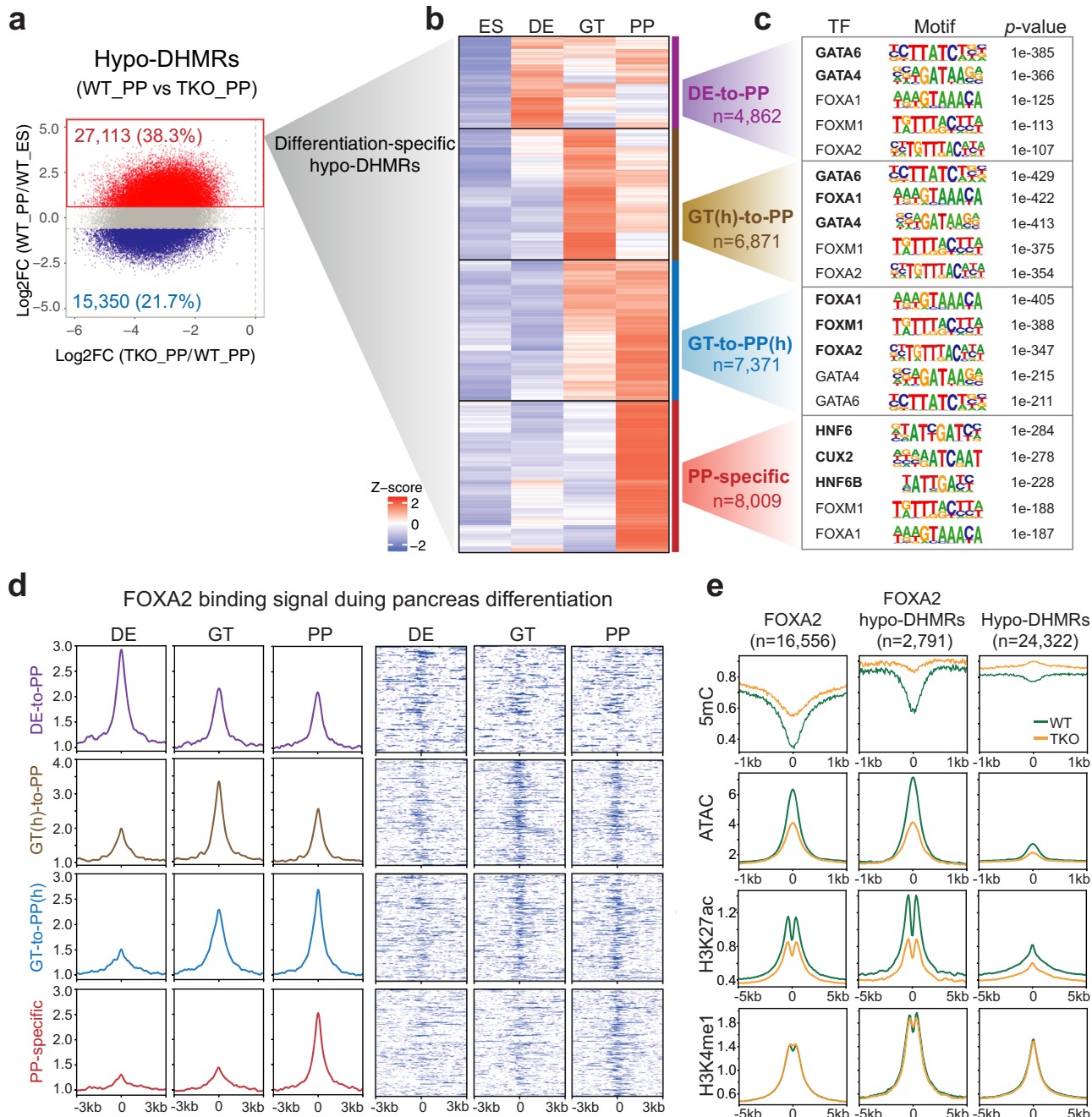

**Fig. 3 Loss of TET disrupts epigenetic dynamics at differentiation-specific hypo-DHMRs associated with distinct pioneer TFs motifs. a** Scatterplot showing a $\log_2$-fold change of 5hmC in pancreatic progenitors (WT_PP) versus hESCs (WT_ES) within hypo-DHMRs identified in TKO_PP cells. Red indicates increased 5hmC and blue shows decreased 5hmC in pancreatic progenitors relative to hESCs (|fold change| ≥ 1.5, FDR < 0.05). **b** Clustering of 5hmC signals in WT cells at ES, DE, GT, and PP stage. Each row represents one hypo-DHMR showing an increased 5hmC in pancreatic progenitors compared with hESCs. Red represents high 5hmC; blue indicates low 5hmC. **c** Motif enrichment analysis among the four clusters defined in **b**. The top five known motifs are shown. The significance was statistically determined by ZOOPS scoring coupled with hypergeometric enrichment calculations without multiple test correction. **d** Average density plots and heatmaps showing dynamic changes of FOXA2 binding in DE, GT, and PP lineage intermediates across the four clusters defined in **b**. The color scale from white to blue in heatmaps represents normalized FOXA2 signal from low to high. **e** Enrichment profile of methylation ratio (5mC/C), chromatin accessibility (ATAC), H3K27ac, and H3K4me1 at FOXA2-binding sites alone (left column), 'differentiation-specific hypo-DHMRs' alone (right column) and 'differentiation-specific hypo-DHMRs' overlapped with FOXA2-binding sites (middle column). The WT signal is marked in green, and the TKO signal is in orange.

active and open chromatin at FOXA2-increased sites in PP (Supplementary Fig. 4c). Although DNA hypermethylation was found in both groups upon TET depletion (Supplementary Fig. 4d), H3K27ac, H3K4me1, and ATAC signals were significantly lost at the FOXA2-decreased but not -increased sites

(Supplementary Fig. 4c). To comprehensively characterize temporal patterns of FOXA2 recruitment at the FOXA2-decreased sites, we clustered FOXA2 binding signal in WT cells at the DE, GT, and PP stages. Three distinct patterns of FOXA2 occupancy were observed. Cluster I regions (149) were bound by

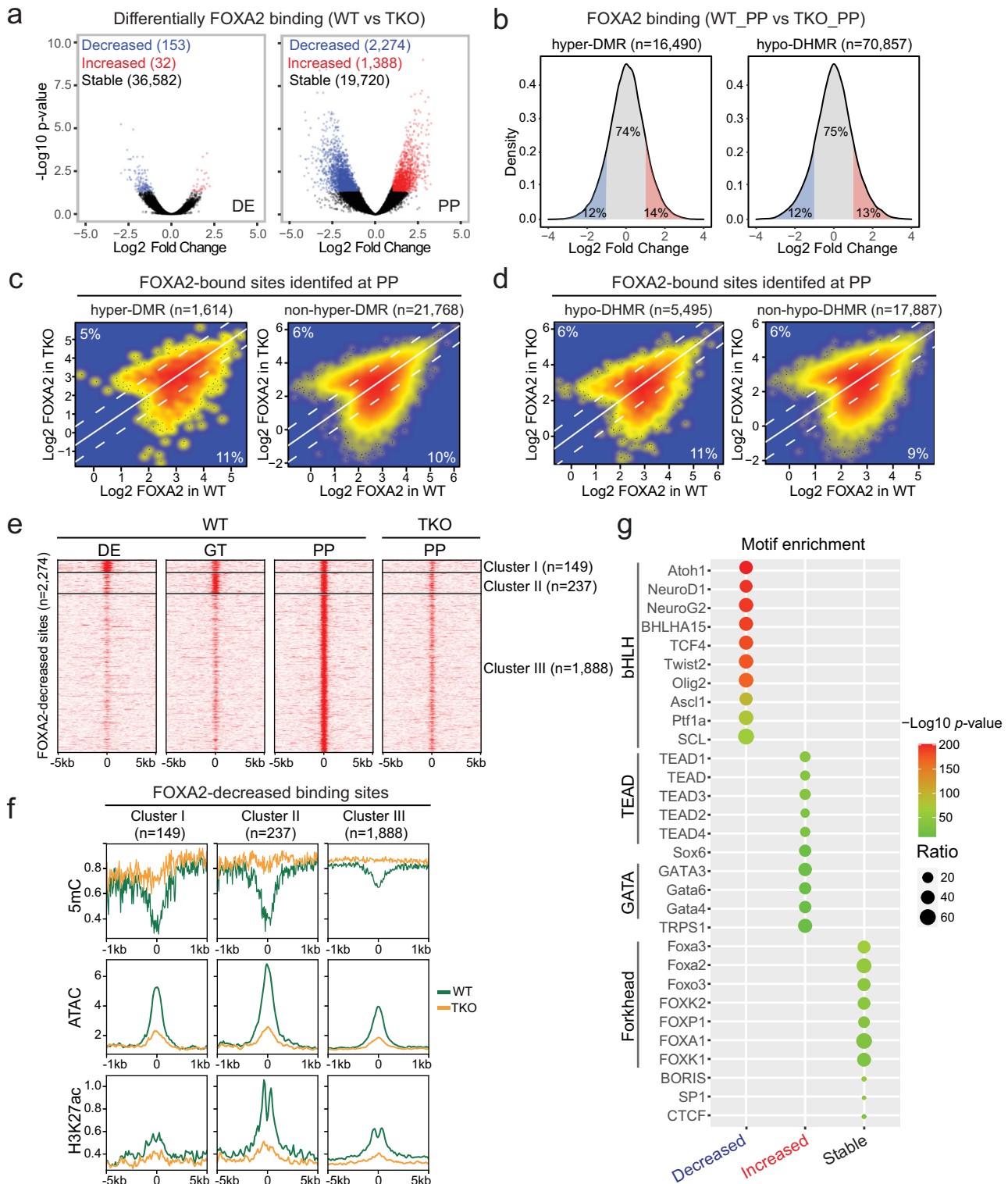

FOXA2 from DE-to-PP stage, cluster II regions (237) were FOXA2-bound at GT and PP stages, while the most predominant group, cluster III (1888), displayed de novo FOXA2 occupancy in pancreatic progenitors (Fig. 4e). We then analyzed the annotations of nearby genes with Genomic Regions Enrichment of Annotations Tool (GREAT) and found that cluster III regions were enriched in terms of neuron and endocrine pancreas development (Supplementary Fig. 4e). Further examination of ATAC-seq, H3K27ac, and 5mC signals in three FOXA2-

decreased clusters revealed high levels of DNA methylation accompanied by low levels of enhancer activity and chromatin accessibility in cluster III regions (Fig. 4f). Taken together, our data suggest that TET-mediated hypomethylation at de novo FOXA2 binding loci provides an integration hub to fine-tune cell fate-specific chromatin activation after pancreas induction.

To identify potential TFs associated with differential FOXA2 binding, we conducted de novo motif analysis to identify unique motifs enriched at FOXA2-decreased sites against a background

**Fig. 4 De novo binding of FOXA2 at differentiation stage-specific genomic loci enriched for bHLH motifs. a** Volcano plots of FOXA2 ChIP-seq reads illustrating differential FOXA2 binding sites identified in TKO cells compared to WT cells at the DE (left) and PP stage (right). Red and blue represent increased and decreased FOXA2 signals in TKO cells, respectively (|fold change| ≥ 2; FDR < 0.05). **b** Density plots showing the percentages of differential FOXA2 binding in TKO_PP compared to WT_PP at hyper-DMRs (left) and hypo-DHMRs (right). The colors red, blue, and gray represents increased, decreased, and non-changed FOXA2 signals in TKO_PP cells, respectively (|fold change| ≥ 2). **c** Scatterplots presenting FOXA2 signals in WT_PP versus TKO_PP at FOXA2 binding sites overlapped with (left) or without (right) hyper-DMRs. Percentages of differential FOXA2 binding are indicated (|fold change| ≥ 2). **d** Scatterplots presenting FOXA2 signals in WT_PP versus TKO_PP at FOXA2 binding sites overlapped with (left) or without (right) hypo-DHMRs. Percentages of differential FOXA2 binding are indicated (|fold change| ≥ 2). **e** Clustering of FOXA2 binding signals at the DE, GT, and PP stage in regions showed lost FOXA2 in TKO_PP cells. **f** Average density plot of methylation ratio (5mC/C), ATAC-seq, and H3K27ac ChIP-seq reads at cluster III FOXA2-decreased regions in PP for WT (green) and TKO (orange) cells. **g** Transcription factor motif enrichment analysis of genomic regions showed decreased, increased, and stable FOXA2 binding in TKO_PP cells. The significance was statistically determined by ZOOPS scoring coupled with hypergeometric enrichment calculations without multiple test correction. The top 10 known motifs are shown. The color represents p-values and the size of circle represents the proportion of peaks containing a TF motif for each group.

of FOXA2-increased and stable sites, and vice versa. Interestingly, loci lost FOXA2 binding, particularly in cluster III, were mostly enriched for basic-helix-loop-helix (bHLH) motifs such as the pancreatic endocrine cell fate determinant NEUROD1[32] (Fig. 4g, Supplementary Fig. 4f, Supplementary Data 9). In contrast, regions gained FOXA2 were mostly enriched with motifs of TEAD and GATA family members, and FOXA2-stable sites feature abundant forkhead motifs (Fig. 4g). Notably, expression of the bHLH TFs including *NEUROD1*, *PTF1A*, and *ASCL1* failed to be induced in TET-knockout cells (Supplementary Fig. 4g) which likely results in a decrease of FOXA2 binding at genomic loci primarily associated with these TFs. In summary, these data imply that de novo recruitment of FOXA2 to differentiation stage-specific sites is genetically and epigenetically primed with the cooperation of additional cell-type-specific TFs.

**TET1 is required in pancreatic β-cell specification**. Our findings suggest that TET inactivation-induced aberrant methylation/hydroxymethylation at least in part contributes to defective β-cell specification. To determine the essential role of each TET family member in pancreatic differentiation, we further analyzed mutants with single (*TET1*, *TET2*, or *TET3*) or double (*TET1/2*, *TET2/3*, or *TET1/3*) TET knockout. Most of the mutant lines were able to induce the expression of *PDX1* and *NKX6.1* to levels comparable to WT except for the TET1/3 knockout line which showed less *NKX6.1* expression at the PE stage (Supplementary Fig. 5a, b). However, only those retaining intact *TET1* expression (TET2KO, TET3KO, and TET2/3DKO) displayed proper induction of *PAX4* (Supplementary Fig. 5c). In comparison with *TET2/TET3* double-deletion, inhibition of TET1 alone had significant effects on the formation of INS- and C-peptide (CPEP)-expressing β-cells but not GCG-expressing α-cells (Fig. 5a, b).

To evaluate global transcriptome changes in response to *TET1* deletion, we performed RNA-seq of TET1KO cells at the PP stage. Integrated analysis of WT_PP, TKO_PP, and TET1KO_PP data revealed a total of 1590 DEGs among the three lines (Supplementary Data 10). Specifically, 555 genes, including most pancreas developmental regulators such as *PDX1*, *SOX9*, *NKX2.2*, *NKX6.1*, *NEUROD1*, and *PAX6*, were downregulated in TKO_PP cells but not in TET1KO_PP cells (Fig. 5c), consistent with effective differentiation of TET1KO-hESCs into PDX1[+] cells (Supplementary Fig. 5a). However, 536 genes, including the β-cell fate determinants *PAX4*, *NKX6.2*, and *FEV*[33–35], were downregulated in TET1-deficient cells, implicating TET1 as responsible for the activation of a subset of genes essential for β-cell identity. To further analyze the functional consequences of TET1 inhibition, WT-, TET1KO-, and TKO-hESCs were differentiated to the PE stage and subsequently engrafted into SCID-beige mice under the kidney capsules (Fig. 5d). Glucose-stimulated human C-peptide secretion was determined 18 weeks post-

transplantation. Notably, mice engrafted with WT_PE cells produced substantial fasting C-peptide in serum (1605 ± 527 pg/ml) and showed statistically significant glucose-stimulated C-peptide secretion (2358 ± 839 pg/ml). By contrast, mice transplanted with TET1KO_PE cells secreted low amounts of basal C-peptide (163 ± 122 pg/ml) and showed no response to glucose stimulation (149 ± 161 pg/ml). In agreement with these functional results, excised WT_PE cell grafts were highly composed of insulin[+] β-cells (Fig. 5e). The TET1KO_PE cell grafts displayed much less insulin content, and only the δ-cell hormone somatostatin was detected in TKO_PE cell grafts. Collectively, these data demonstrate that loss of TET1 impairs β-cell specification and maturation.

**Full-length TET1 is required for *PAX4* enhancer hypomethylation**. To further determine whether TET1 is responsible for β-cell specification, we restored the full-length (*TET1FL*) as well as the *TET1*-catalytic-domain (*TET1CD*) expression, respectively, using lentivirus-mediated gene transduction in TKO cells. In contrast to the catalytically inactive *TET1* (*TET1CD^mut*) (H1672Y, D1674A), both *TET1FL*- and *TET1CD*-transduced cells showed a global increase in 5hmC levels, confirming the replenishment of hydroxymethylation (Supplementary Fig. 6a). Differentiation of the TKO-TET1FL line toward pancreatic endocrine fate led to robust induction of β-cell determinants, including *PDX1*, *NKX6.1*, and *PAX4*, whereas transduction of *TET1CD* only resulted in a slight increase of *NKX6.1* and *PAX4* (Fig. 6a, b, Supplementary Fig. 6c). FACS analysis and immunostaining further demonstrated that TET1CD-transduced cells showed an effective generation of GCG-producing α-cells but not INS/CPEP-expressing β-cells at the PP stage (Supplementary Fig. 6b, c), indicating that full-length TET1 is critical for β-cell specification.

As TET1 depletion inhibited the induction of genes essential for β-cell specification (Fig. 5), we reasoned that co-binding of TET1 and FOXA2 at distal regulatory elements modulates focal methylation status and subsequent gene activation. To identify potential interaction between TET1 and FOXA2, we performed co-immunoprecipitation in TKO-TET1FL cells expressing a FLAG-tagged *TET1FL* gene. Anti-FLAG antibodies co-precipitated endogenous FOXA2 in TKO-TET1FL cells differentiated at the DE stage (Fig. 6c). Reciprocally, co-immunoprecipitation using an anti-FOXA2 antibody led to successful pulldown of ectopic TET1 proteins (Fig. 6c). To exclude the possibility of DNA-protein interaction mediated co-immunoprecipitation, we treated the cell extracts with nuclease and performed co-immunoprecipitation in the presence of nuclease (Fig. 6c). Our results demonstrated that nuclease treatment did not result in an appreciable decrease of TET1 proteins pulled down by FOXA2, suggesting a physical interaction between FOXA2 and TET1. Although other TFs might

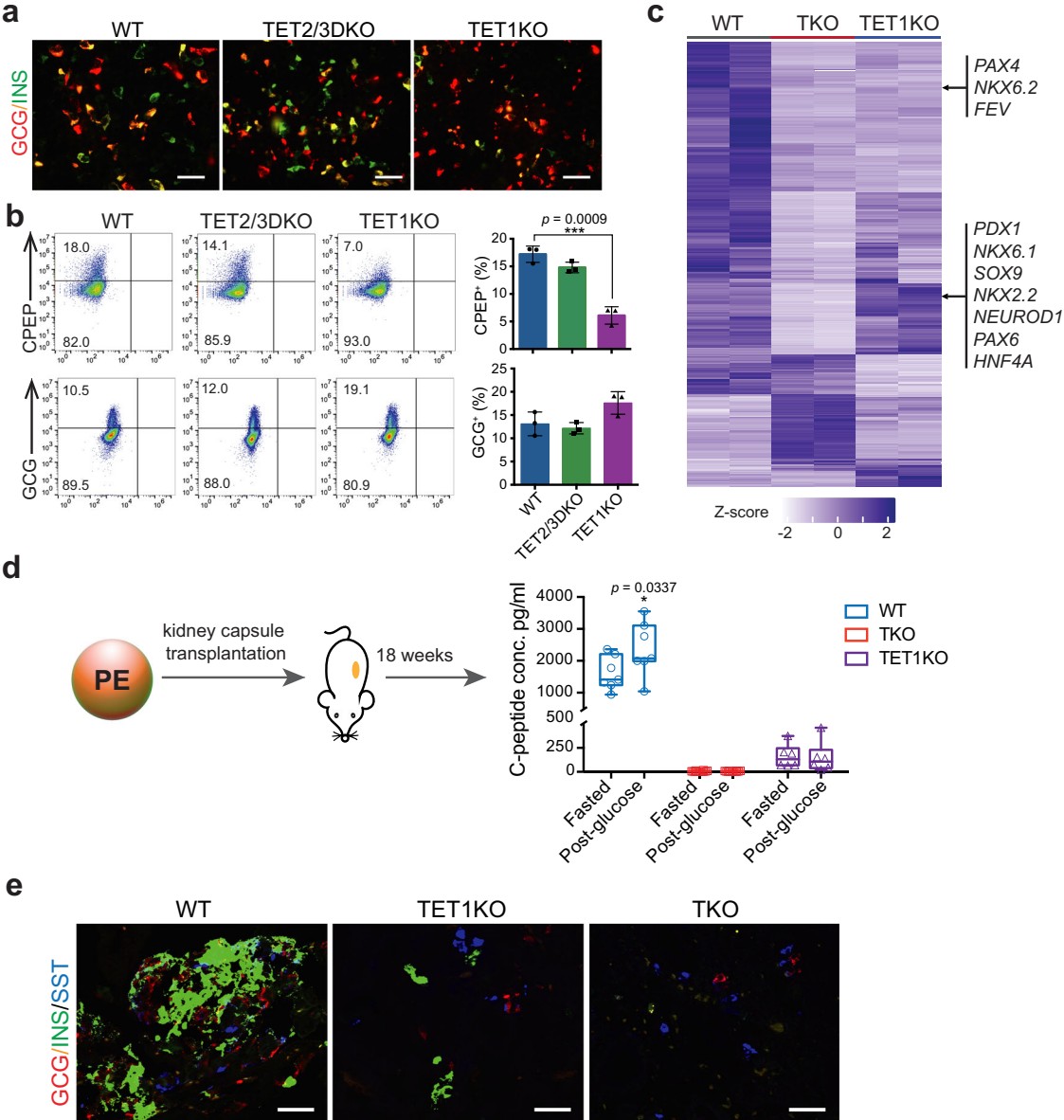

**Fig. 5 TET1-knockout cells show impaired differentiation into functional β-cells. a** Immunostaining of INS, and GCG in WT, TET2/3DKO, and TET1KO cells at the PE stage ($n = 3$ independent differentiation; scale bar = 50 μm). **b** Representative plots of flow cytometry of human C-peptide (CPEP) and glucagon (GCG) in WT, TET2/3DKO, and TET1KO cells at the PE stage. Quantifications of the percentage of CPEP⁺ or GCG⁺ cells are shown in the right panel. ($n = 3$ independent differentiations; student's $t$-test, 2-sided; without multiple test correction). All bar graphs show mean ± SD. **c** Heatmap showing the hierarchical clustering of DEGs among WT, TKO, and TET1KO cells at the PP stage. Each row represents one DEG, and each column represents one biological replicate. The color scale from white to blue represents normalized gene expression levels from low to high (|fold change| ≥ 2; FDR < 0.05). **d** WT ($n = 7$), TKO ($n = 12$), and TET1KO ($n = 6$) cells were differentiated to the PE stage and transplanted under the kidney capsule of nondiabetic female SCID-beige mice. Eighteen weeks post-implantation, human C-peptide levels were measured after an overnight fast and 30 min following an i.p. glucose injection. C-peptide levels from individual mice are shown in box and whisker plots (Plots are centered on mean, with box encompassing 25th-75th percentile and whiskers representing minimum to maximum range; student's $t$-test, 1-sided; without multiple test correction). **e** Immunostaining of insulin (INS), glucagon (GCG), and somatostatin (SST) in cell grafts from WT_PE, TET1KO_PE, and TKO_PE transplanted mice ($n = 2$ independent experiments; scale bar = 50 μm).

act in addition to FOXA2 to promote TET1 deposition, our data illustrate that FOXA2 contributes to TET1 recruitment at FOXA2-binding loci to mediated demethylation during lineage induction.

Since the induction of *PAX4* was inhibited in TET1-deficient lines (Supplementary Fig. 5b), we asked if TET1 would cooperate with FOXA2 to mediate focal demethylation at the *PAX4* enhancer. We found a putative *PAX4* enhancer (~4.0 kb upstream of the TSS), where FOXA2 bound and displayed hypermethylation in TET-deficient cells (Fig. 6d, pink area). We next examined

whether the elimination of TET1 alone results in hypermethylation at this site. Locus-specific methylation was determined using a glucosylation and digestion-based method followed by qPCR analysis. Two independent pairs of primers (PAX4-P1 and PAX4-P2) located within the FOXA2-binding site (Fig. 6d) were used to determine the percentage of methylated and non-methylated cytosine in TET1KO, TET2/3DKO, TKO, and WT cells at the PP stage. Notably, the percentage of 5mC was increased to 35-50% for TET1KO and 60-65% for TKO compared with WT but

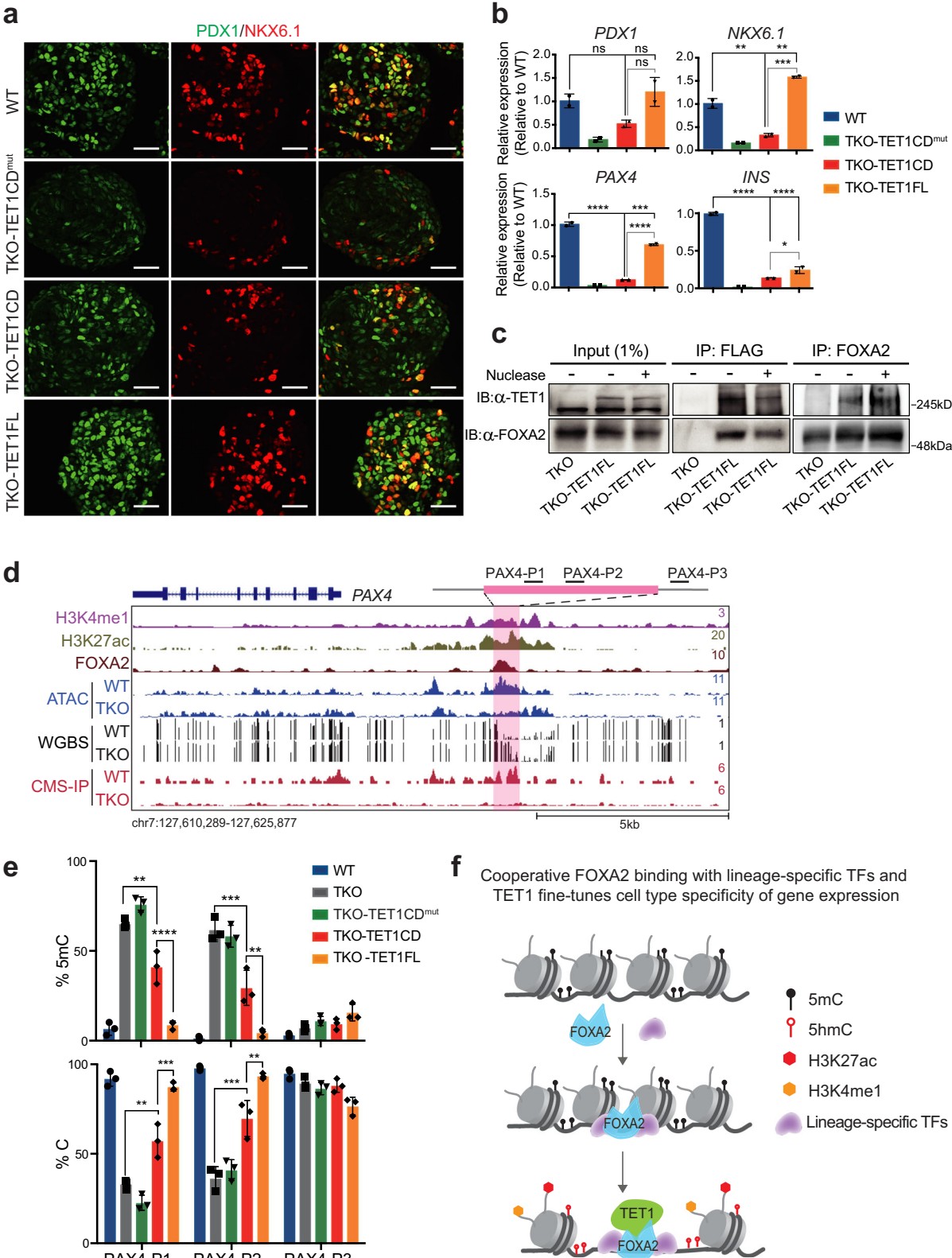

remained unchanged for TET2/3DKO (Supplementary Fig. 6d, left panel). A corresponding decrease in unmethylated cytosine content was observed, with values of 45-65% and ~35% of C in TET1KO and TKO samples, respectively (Supplementary Fig. 6d, right panel). As a housekeeping control, we examined methylation content at a nearby region using the PAX4-P3 primer, and as expected, no significant differences in 5mC and C were observed

in any samples. We then determined methylation contents at the same *PAX4* locus in TET1-replenished TKO cells. Consistent with the results found in TET2/3DKO cells, no detectable differences in 5mC and C were found between TKO-TET1FL and WT samples (Fig. 6e). However, 5mC levels were significantly higher with a value of 30-40% in TKO-TET1CD than TKO-TET1FL samples (Fig. 6e, top panel). Together with the

**Fig. 6 Full-length TET1 is required for the establishment of a hypomethylated PAX4 enhancer. a** Immunostaining of PDX1 and NKX6.1 at the PP stage for WT, TKO-TET1CD$^{mut}$, TKO-TET1CD, and TKO-TET1FL cells ($n = 3$ independent differentiations; scale bar = 50 µm). **b** Expression of *PDX1*, *NKX6.1*, *PAX4*, and *INS* in WT, WT, TKO-TET1CD$^{mut}$, TKO-TET1CD, and TKO-TET1FL cells at the PE stage ($n = 2$ independent differentiations; $p = 0.001$, $p = 0.0021$, $p = 0.0001$, $p = 7.4 \times 10^{-7}$, $p = 0.0002$, $p = 3.4 \times 10^{-5}$, $p = 1.8 \times 10^{-5}$, $p = 3.7 \times 10^{-5}$, and $p = 0.0498$ for WT versus TKO-TET1CD, WT versus TKO-TET1FL, and TKO-TET1CD versus TKO-TET1FL of relative expression for *NKX6.1*, *PAX4*, and *INS*, respectively; one-way ANOVA with Turkey's multiple comparison test). All bar graphs show mean ± SD. **c** Immunoprecipitation of ectopic TET1 or endogenous FOXA2 in TKO-TET1FL cells. Cells were subjected to immunoprecipitation using anti-FLAG or anti-FOXA2 antibodies, followed by immunoblotting of TET1 or FOXA2. Nuclease treatment on TET1-FOXA2 interaction is indicated. TKO is included as a control ($n = 3$ independent experiments). **d** Genome-browser view of the *PAX4* locus with increased methylation and decreased chromatin accessibility upon TET depletion at a FOXA2-bound site featured enhancer signatures H3K4me1 and H3K27ac. **e** Locus-specific increase in 5mC at the *PAX4* enhancer in TKO or TKO-TET1CD samples compared with WT and TKO-TET1FL samples. Percentages of unmethylated cytosine and 5mC at CCGG sites are shown ($n = 3$ independent differentiations; $p = 0.0011$, $p = 9.1 \times 10^{-5}$, $p = 0.0005$, $p = 0.0033$, $p = 0.0014$, $p = 0.0003$, $p = 0.0004$, and $p = 0.0055$ for TKO versus TKO-TET1CD and TKO-TET1CD versus TKO-TET1FL of percentages of 5mC and unmethylated cytosine at *PAX4-P1* and *PXA4-P2* loci, respectively; one-way ANOVA with Turkey's multiple comparison test). All bar graphs show mean ± SD. **f** Schematic model depicting cooperative FOXA2 recruitment with lineage-specific TFs and TETs. Lineage-specific TFs, such as PTF1A and NEUROD1, stabilize the binding of FOXA2 which enables the recruitment of TET1 to induce DNA demethylation, chromatin opening, and fine-tuning cell type-specific gene expression.

insufficient restoration of *PAX4* in TKO-TET1CD cells (Fig. 6b), we conclude that full-length TET1 is responsible for the establishment or maintenance of a hypomethylated state at the *PAX4* enhancer, which is essential for transcription activation of *PAX4* and subsequent β-cells generation.

## Discussion

In the present work, we dissected the roles of TET proteins in pancreatic endocrine commitment based on a stepwise hESC differentiation system. We found that the loss of all three TET family members significantly impaired the differentiation of pancreatic β-cells. Furthermore, we discovered that locus-specific hypermethylation was associated with genes essential for β-cell specification and maturation, such as *PAX4*[33], *PDX1*[36], and *NKX2.2*[37]. The reintroduction of *TET1* in TET-deficient cells effectively reversed hypermethylation and restored the expression of *PAX4*. We further demonstrated that TET1 functions as an upstream epigenetic regulator of *PAX4* presumably through direct recruitment by FOXA2 to a putative *PAX4* enhancer to preserve its unmethylated status, thereby potentiating *PAX4* expression to adopt β-cell fate during endocrine lineage commitment. Consistently, we observed striking increases in methylation at the *PAX4* enhancer in TET1-knockout cells but not in TET2/TET3 DKO cells, suggesting that TET1 epigenetically regulates induction of the β-cell program in a locus-specific manner. Moreover, despite the successful induction of *PDX1* and *INS* upon deletion of TET1 alone, mice receiving TET1KO cell grafts showed a persistent defective insulin response to glucose, implying that TET1 is also essential for β-cell maturation.

Whereas the specification of β-cells was strongly influenced by depletion of TET1, we did not observe a significant inhibition on the expression of *ARX*, a critical α-cell fate determinant[38]. In contrast to *PAX4*, no hyper-DMRs were identified within the *ARX* locus, where methylation levels were nearly undetectable in pancreatic progenitors. Previous studies demonstrate that several CpG-rich sites of *Arx*, including one site close to TSS and another site 2 kb upstream of TSS, are heavily methylated in adult β-cells but not α-cells[39]. Moreover, pharmacological inhibition of DNA methyltransferases, *Dnmts*, in pancreatic progenitors promotes α-cell specification[40], whereas deletion of *Dnmt1* in β-cells results in demethylation and depression of *Arx*[39]. We thus speculate that *ARX* is hypomethylated in pancreatic progenitors in a TET-independent manner. The *ARX* locus maintains an unmethylated state when progenitors differentiate into α-cells, whereas it becomes hypermethylated once cells commit to β-cell fate in the presence of DNMTs and other β cell-specific regulators. In line with this hypothesis are findings that Nkx2.2 recruits Dnmt3a to

the *Arx* promoter to repress its expression in β-cells[41]. In the future, it will be interesting to explore whether there are differences in the complete epigenetic landscapes of endocrine progenitors, which subsequently differentiate into α- or β-cells.

Interestingly, even TET-mediated 5hmC was dynamically enriched at the definitive endoderm stage (Fig. 3b), we did not find that loss of TETs impaired endoderm formation. In line with our findings, previous studies have revealed that depletion of TETs in mouse ESCs inhibits adoption of neural cell fate while concomitantly skewing toward mesoderm lineage[17,42]. Indeed, it has been recently demonstrated that Nodal signaling, which promotes endoderm and mesoderm differentiation[43,44], is hyperactivated in TET-deficient mouse embryos[45] suggesting activation of Nodal signaling sustains endoderm formation upon TET depletion.

TET dioxygenases are critical for lineage induction in a cell-type-specific manner[46]. How TETs recognize lineage-specific regulatory elements and modulate chromatin remodeling during pancreas development remains unknown. Here, we addressed these questions by performing analyses integrating gene expression with multiple chromatin features, including DNA methylation, hydroxymethylation, chromatin accessibility, and histone modifications of enhancers. We found extensive hypermethylation in TET-deficient cells that differentiated to the pancreatic progenitor stage. A significant portion of hyper-DMRs was distributed in a differentiation-specific manner, in which they were enriched for both pioneer and lineage-specific TFs and showed remarkable decreases in chromatin activity upon TET inactivation. Given that pioneer TFs are core components of the transcriptional complexes at *cis*-regulatory elements during differentiation, our data suggest that TETs are essential for enhancers activation and subsequent incorporation of lineage-specific TFs.

During lineage specification, chromatin structure is dynamically changed between 'closed' and 'open' states. Open chromatin regions such as primed and active enhancers are accessible for lineage-specific TF and epigenetic modulator binding to initiate gene transcription. Proper chromatin remodeling is believed to be at least partly triggered by pioneer TFs, which can directly bind to nucleosomal DNA[47]. Previous studies have shown that pioneer TF FOXA initiates enhancer priming upon lineage induction[3,4]. A recent study has further demonstrated that secondary recruitment of FOXA to unprimed enhancers at the late stage of lineage induction is required for enhancer activation[21]. Our analyses revealed that (1) changes in 5hmC mirrored the dynamic binding of FOXA2 in cells differentiated from hESCs through defined lineage intermediates toward pancreatic endocrine fate, (2) upon

TET depletion chromatin activity was markedly decreased at hypo-hydroxymethylated regions associated with de novo FOXA2 recruitment, and (3) FOXA2 physically interacted with TET1, suggesting that FOXA2 favors TET1 deposition to facilitate local chromatin remodeling at the late stage of pancreas induction, while de novo recruitment of FOXA2/TET1 fine-tunes gene induction in the pancreatic progenitor domain. Hence, the identification of preferential enrichment of TET1 in a FOXA2-dependent manner will further strengthen the specificity of the TET1-FOXA2 axis during lineage specification.

How pioneer TFs selectively recognize differentiation stage-specific target sites and mediate chromatin remodeling is still mostly unknown. Our present study demonstrates that (1) de novo FOXA2 recruitment occurs at genomic loci with low levels of active chromatin modifications dependent on TETs, (2) FOXA2-stable sites are predominantly abundant for the forkhead family motifs, whereas (3) FOXA2-decreased sites harbor motifs for bHLH TFs, such as NEUROD1 and PTF1A, which fail to be induced in TET-deficient cells, implying a subset of FOXA2 recruitment-associated chromatin activation requires lineage-specific TFs (Fig. 6f). Consistent with this possibility, it has been recently suggested that stronger and more abundant FOXA motifs are enriched at primed enhancers where FOXA1/2 binding is independent of lineage-specific TFs, whereas fewer and more degenerate FOXA motifs are enriched at unprimed enhancers where de novo FOXA1/2 recruitment is dependent on pancreatic lineage-determining TF PDX1[21]. Although it has been reported that FOXA is necessary for chromatin remodeling for both primed and unprimed enhancers during pancreatic lineage induction, the precise mechanism by which epigenetic modulators participate in these processes is unclear. Our findings provide molecular insights into lineage induction and cell-type determination during organogenesis. Our results support that de novo recruitment of FOXA2 to distal regulatory elements favors TET proteins' local deposition to regulate chromatin remodeling and cell fate decisions. We postulate that lineage-specific TF-dependent recruitment of FOXA2/TET1 lowers the threshold of cell-type-specific gene expression, and therefore fine-tunes the activation of a specific lineage program during developmental transition. Further investigation is warranted to determine how TFs stable FOXA2 binding and synergize the recruitment of multiple chromatin remodeling machineries.

## Methods

**Cell lines**. All experimental protocols were approved by the University of Macau Panel of Research Ethics. hESC research was carried out in accordance with the National Institutes of Health Guidelines on Human Stem Cell Research. Experiments involving laboratory animals were approved by the University of Macau Animal Research Ethics Committee (protocol UMARE-042-2020). The human embryonic stem cell line H1 was obtained from WiCell Research Institute. H1 cells were maintained on Matrigel (Corning) in mTeSR1 (STEMCELL Technologies) and passaged every 4-5 days using 0.5 mM EDTA and 3 mM NaCl. Cells were tested negative for mycoplasma by ELISA of cell culture supernatants. HEK293T cells were purchased from ATCC and maintained in Dulbecco's Modified Eagle Medium (Gibco) supplemented with 10% fetal bovine serum (FBS). All cells were cultured in a humidified incubator at 37 °C with 5% $CO_2$.

**Pancreatic differentiation**. All differentiation experiments were repeated at least three times with three individual clones of the same phenotype. Pancreatic differentiation was performed as previously described with slight modifications[48]. In brief, hESCs were dissociated with Accutase (eBioscience) and seeded at a density of 19,000 cells/cm² on Matrigel in mTeSR1 supplemented with 10 μM Rho-associated protein kinase inhibitor Y-27632 (Miltenyi Biotec). Upon reaching 95% confluence, cells were exposed to differentiation medium with daily media feeding. Alternatively, a suspension-based differentiation format was used as previously described[25]. Briefly, $5.5 \times 10^6$ cells were suspended in 5.5 ml mTeSR1 per well of 6-well ultra-low attachment plates (Corning Costar) and cultured overnight in an orbital shaking $CO_2$ incubator (Eppendorf New Brunswick) at 95 rpm. Cell aggregates were supplied with differentiation medium and continually rotated at 95 rpm with daily media feeding.

ES-to-DE (3 d): hESCs were exposed to 100 ng/ml Activin A (PeproTech) and 25 ng/ml Wnt-3a (PeproTech) in basal medium-I containing MCDB 131 medium (Gibco), 1.5 g/l $NaHCO_3$, 1× GlutaMAX (Gibco), 10 mM glucose, and 0.5% BSA for 1 day. For two additional days, cells were cultured in basal medium-I further supplemented with 100 ng/ml Activin A.

DE-to-GT (3 d): DE intermediates were incubated in basal medium-I supplemented with 50 ng/ml FGF7 (PeproTech) for 3 days.

GT-to-PP (5 d): GT intermediates were cultured for 3 days in basal medium-II containing MCDB 131 medium, 2.5 g/l $NaHCO_3$, 1× GlutaMAX, 10 mM glucose, 2% BSA, and 1:200 ITS-X (Gibco), which was further supplemented with 50 ng/ml FGF7, 0.25 μM hedgehog inhibitor SANT-1 (Sigma), 1 μM retinoic acid (Sigma), 100 nM BMP inhibitor LDN193189 (Stemgent), and 200 nM PKC activator TPB (Millipore). After 3 days of culture, cells were treated for 2 days with 2 ng/ml FGF7, 0.25 μM SANT-1, 0.1 μM retinoic acid, 200 nM LDN193189, and 100 nM TPB in basal medium-II.

PP-to-PE (3 d): PP intermediates were differentiated in basal medium-III containing MCDB 131 medium, 1.5 g/l $NaHCO_3$, 1× GlutaMAX, 20 mM glucose, 2% BSA, and 1:200 ITS-X, which was further supplemented with 0.25 μM SANT-1, 0.05 μM retinoic acid, 100 nM LDN193189, 1 μM T3 (3,3',5-Triiodo-L-thyronine sodium salt, Sigma), 10 μM ALK5 inhibitor II (Enzo Life Sciences), 10 μM $ZnSO_4$, and 10 μg/ml heparin (Sigma).

**Animal experiments**. Immunocompromised SCID-beige mice were obtained from Charles River. Mice (6-week-old, both sexes) were maintained at 22 ± 1 °C and 40-60% humanity under a 12-h light/dark cycle with free access to normal chow food and water. For transplantation, d14 cell aggregates were further incubated in basal medium-III supplemented with 100 nM LDN193189, 1 μM T3, 10 μM ALK5 inhibitor II, 10 μM $ZnSO_4$, and 100 nM γ-secretase inhibitor XX (Millipore) for 1 day. Cell aggregates (~$5 \times 10^6$ cells) were then transplanted into 6-week-old SCID-beige mice under the kidney capsule as previously described[49].

Glucose-stimulated human C-peptide secretion was assessed with mice 18 weeks post-transplantation. Blood samples were collected after overnight fasting and 30 min following an intraperitoneal injection of glucose (2 g/kg body weight). Human C-peptide levels in isolated plasma were quantified using the STELLUX Chemi Human C-peptide ELISA kit (ALPCO Diagnostics) according to the manufacturer's instructions.

**Generation of TET-knockout lines**. TET-knockout hESCs were generated using CRISPR/Cas9 technology. gRNAs (Supplementary Table 1) were designed to target the sequences encoding exon 7 of TET1, exon 3 of TET2, or exon 3 of TET3 and cloned into pX330 vector (Addgene #42230) as previously described[50]. Constructs containing validated gRNAs were electroporated together with a vector expressing puromycin into hESCs using the P3 Primary Cell 4D-Nucleofector X kit (Lonza) following the manufacturer's instructions. The electroporated cells were plated at ~2000 cells/cm² and cultured on Matrigel in mTeSR1 supplemented with 10 μM Y-27632 for 2 days. Successfully transfected cells were selected with 1 μg/ml puromycin in mTeSR™1 and allowed to expand to form visible colonies from a single cell. Subsequently, clonal colonies were manually picked and reseeded individually into 24-well plates. The amplified colonies were analyzed by Sanger sequencing at targeted loci for the presence of Indel mutations.

**TET1 overexpression cell lines**. To generate overexpression constructs for TET1, the C-terminal FLAG-tagged human full-length TET1 (TET1FL), TET1 catalytic domain (TET1CD), and TET1 catalytic inactive mutant (TET1CD^mut) were amplified from FH-TET1-pEF (Addgene #49792) or pIRES-hrGFP II-mTET1 (Addgene #83569) and cloned into the NotI digested PCDH-CAG-MCS-P2A-Puro vector (a kind gift from Dr. R. Xu, University of Macau)[51]. Lentivirus particles were prepared as previously described[52]. For viral transduction, TKO-hESCs were grown on Matrigel in mTeSR1 using 24-well plates and treated with 6 μg/ml polybrene for 15 min at 37 °C. Concentrated lentivirus particles (10 μl, $1 \times 10^6$ TU/ml) were added to cell culture and incubated overnight at 37 °C. On the next day, viral infection was repeated (30 μl, $1 \times 10^6$ TU/ml) to increase transduction efficiency. Infected cells were cultured in mTeSR1 for 2 days and then exposed to 1 μg/ml puromycin for 10 days. The TET1 overexpression cell lines TKO-TET1FL, TKO-TET1CD, TKO-TET1CD^mut, were amplified and frozen down.

**Co-immunoprecipitation**. Cells were lysed in ice-cold lysis buffer (20 mM Tris pH 7.5, 150 mM NaCl, 1 mM EDTA, 1% Triton X-100, 5 mM 4-nitrophenyl phosphate di(Tris) salt, 2 mM $Na_2VO_4$, 0.5% sodium deoxycholate, protease inhibitors), sonicated in Bioruptor® Plus (Diagenode) for 5 min (30 s on, 30 s off), and then centrifuged at 14,000 × g for 10 min at 4 °C. The supernatant containing 4 mg of cell lysate was treated with 20 U nuclease (Takara) at 4 °C for 30 min and then incubated with 5 μg mouse anti-FLAG (Sigma F1804) or 5 μg goat anti-FOXA2 (R&D AF2400) antibody overnight at 4 °C with gentle rocking, followed by incubation with 25 μl protein A/G magnetic beads (Thermo Fisher Scientific) at 4 °C for 4 h. The beads were washed three times with wash buffer I (20 mM Tris pH 8.0, 0.3 M KCl, 10% glycerol, 1 mM EDTA, 1 mM DTT, 0.1% NP-40, and proteinase inhibitors) and two times with wash buffer II (20 mM Tris pH 8.0, 0.1 M KCl, 10% glycerol, 1 mM EDTA, 1 mM DTT and proteinase inhibitors). 4× Laemmli sample

buffer with β-mercaptoethanol was added to washed beads, and then boiled for 10 min at 95 °C for western blotting analyses.

**Western blot analysis**. Immunoprecipitated samples were separated using 8% SDS-PAGE and transferred onto polyvinylidene difluoride (PVDF) membranes. The membranes were blocked with 5% non-fat skim milk at room temperature for 1 h and then incubated with the relevant primary antibodies overnight at 4 °C, followed by incubation with HRP-conjugated secondary antibodies at room temperature for 1 h. Antibodies used were goat anti-FOXA2 (R&D AF2400) and rabbit anti-TET1 (GeneTex GTX124207). The blots were detected by ECL detection reagents (Thermo Fisher Scientific) and captured using ChemiDoc Imaging System (Bio-Rad). Antibody dilutions used for western blot are listed in Supplementary Table 2.

**Locus-specific detection of 5mC**. Detection of 5mC content at particular CCGG sites was performed using the Epimark 5hmC and 5mC analysis kit (New England Biolabs) following the manufacturer's instructions. Briefly, genomic DNA was extracted using the DNeasy Blood and Tissue kit (Qiagen) followed by RNase treatment. RNase-treated DNA samples were incubated with T4 β-glucosyltransferase at 37 °C for 16 h. The glycosylated DNA was subsequently digested with *MspI* or *HpaII* for 8 h at 37 °C. Samples were treated with Proteinase K at 40 °C for 30 min and then at 95 °C for 10 min to inactivate the enzymes. Site-specific methylation contents were examined by RT-qPCR using the primers listed in Supplementary Table 3. The percentage of 5mC and unmodified cytosine were calculated using the comparative $C_t$ method.

**5hmC dot blot assay**. Genomic DNA was extracted using the DNeasy Blood and Tissue kit (Qiagen) according to the manufacturer's instructions. DNA was denatured in 1 M NaOH supplemented with 25 mM EDTA at 95 °C for 10 min and then neutralized with 2 M ice-cold ammonium acetate for 10 min. Two-fold serial dilutions of the DNA samples were spotted onto the nitrocellulose membrane. The air-dried membrane was fixed with UV irradiation (CL-1000 UV crosslinker, Ultra-Violet Products), blocked with 5% non-fat skim milk, and incubated with a rabbit anti-5hmC antibody (Active Motif 39769) followed by an HRP-conjugated anti-rabbit antibody (1:5000, Jackson ImmunoResearch). Signal was visualized with SuperSignal West Pico PLUS chemiluminescent substrate (Thermo Fisher Scientific). The same membrane was subsequently stained with 0.02% methylene blue in 0.3 M sodium acetate to ensure equal loading of input DNA.

**RNA isolation for real-time quantitative PCR**. Total RNA was extracted using the RNeasy Plus Mini kit (Qiagen) according to the manufacturer's instructions. cDNA was synthesized using the PrimeScript RT reagent kit (Takara). Real-time quantitative PCR was performed in triplicate using the SYBR Premix Ex Taq (Tli RNase H Plus) kit (Takara). The expression of *TBP* was used for the normalization of mRNA expression. All primers used for RT-qPCR were listed in Supplementary Table 3.

**Immunocytochemical analysis**. Differentiated cells were fixed in 4% paraformaldehyde for 30 min at room temperature, washed, and then permeabilized with 0.15% Triton X-100 at room temperature for 1 h. Following blocking with 5% normal donkey serum (Jackson Immuno Research Laboratories), samples were incubated with primary antibodies at 4 °C overnight and then appropriate secondary antibodies for 1 h at room temperature. Antibodies used were rabbit anti-OCT4 (Cell signaling 2840); goat anti-FOXA2 (R&D AF2400); rabbit anti-somatostatin (Abcam ab64053); guinea pig anti-insulin (DAKO A0564); mouse anti-glucagon (Sigma G2654); goat-anti-PDX1 (Abcam ab47383); and mouse anti-NKX6.1 (DSHB F55A10). Images were acquired using the Zeiss Axio Observer microscope. For transplant grafts, tissues were fixed with 4% paraformaldehyde overnight at 4 °C, washed with PBS, and subsequently exposed to 30% sucrose overnight at 4 °C. Samples were mounted with Optimal Cutting Temperature Compound (Tissue-Tek) and sectioned at 10 μm. Immunofluorescent staining was performed on cryosections as described above. Antibody dilutions used for immunofluorescent staining are listed in Supplementary Table 2.

**Flow cytometry analysis**. Cells derived from hESCs were incubated with Accutase® at room temperature to obtain a single-cell suspension. Cells were washed with ice-cold buffer compromised 0.2% BSA in PBS and fixed with 4% paraformaldehyde for 20 min at 4 °C. Fixed cells were permeabilized with 1 × BD Perm/Wash Buffer™ (BD Biosciences) and stained with primary antibodies diluted in 1× BD Perm/Wash Buffer™ for 1 h at 4 °C. Antibodies used were mouse anti-Oct3/4-Alexa Fluor 647 (BD Biosciences 560329); mouse anti-SOX17-PE (BD Biosciences 561591); goat anti-PDX1 (R&D AF2419); mouse anti-glucagon (Sigma G2654); rat anti-C-peptide (DSHB GN-ID4); and mouse anti-NKX6.1 (DSHB F55A12). Cells were subsequently washed, stained with appropriate secondary antibodies for 1 h at 4 °C, and assessed using an Accuri C6 flow cytometer (BD Biosciences). Data were analyzed using FlowJo software (Tree Star).

**RNA-seq library construction and data analysis**. The integrity of extracted RNA was determined using an Agilent 2100 Bioanalyzer (Agilent Technologies). Subsequently, polyA-tailed RNA was selected using Dynabeads oligo (dT) (Thermo Fisher Scientific), and libraries were prepared using the NEBNext Ultra RNA Library Prep kit for Illumina (New England Biolabs). Libraries were subjected to high-throughput sequencing on an Illumina Novaseq 6000 system (150 cycles, paired-end) at Novogene (Tianjin, China). To process the sequencing data, low-quality bases and adapter were trimmed using TrimGalore v0.5.0 (options: --quality 20 and --length 50, https://github.com/FelixKrueger/TrimGalore). Clean reads were aligned to the hg38 genome reference using STAR v2.5.3[53] with default parameters, and only uniquely mapped reads were used for downstream analysis. A count matrix for each gene was generated using htseq-count (HTSeq package[54]). DESeq2 v1.28.1[55] was used to identify significant DEGs in knockout samples compared with WT samples at different differentiation stages (|fold change| ≥ 2; False Discovery Rate (FDR) < 0.05). Hierarchical cluster analysis of the union DEGs was used to determine stage-specific signature genes. The 'ClusterProfiler' package in R was used for the functional enrichment analysis of DEGs in KEGG pathways. BAM files were converted to bigwig files by Bam2wig.py in RSeQC[56–58].

**ATAC-seq library preparation and data analysis**. ATAC-seq libraries were prepared as previously described[59]. In brief, cells were enzymatically dissociated and lysed in lysis buffer (10 mM Tris-HCl, pH 7.4, 10 mM NaCl, 3 mM MgCl2, 0.1% IGEPAL CA-630). Immediately after centrifugation, transposition reactions were carried out by adding Tn5 transposes from the Illumina Nextera DNA library preparation kit to the isolated nuclei and incubation at 37 °C for 30 min. DNA fragments were purified using the MinElute PCR Purification kit (Qiagen) and amplified using the KAPA real-time library amplification kit (Roche). Libraries were purified using PCRClean DX beads (Aline Biosciences) and subsequently subjected to high-throughput sequencing on an Illumina NextSeq 500 instrument (75 cycles, paired-end).

Adapter trimming of raw reads was performed using TrimGalore v0.5.0 (options: --quality 20 and --length 20), and high-quality (Q ≥ 20) reads were uniquely aligned to the hg38 genome reference using Bowtie2 with the '--very-sensitive' option. Reads mapped to mitochondrial DNA and PCR duplicate reads were removed, and uniquely mapped reads were extracted for downstream analysis. Genrich v0.5 (https://github.com/jsh58/Genrich) with ATAC-seq mode (option: -j, -q 0.01) was applied for each sample (with two biological replicates) to call ATAC-seq peaks. In total, 52,817 and 38,697 peaks were detected in WT_PP and TKO_PP cells, respectively. BEDTools[60] intersect was used to generate a total of 124,322 non-overlapping peak regions, and DeepTools multiBamSummary was applied to count the reads falling into peak regions. The counts were used in DESeq2 v1.28.1 for normalization and identification of significant differential accessible regions (DARs) between WT_PP and TKO_PP cells by the criteria (|fold change| ≥ 2; FDR < 0.05). Volcano plots were generated using the R package ggplot2[61]. Motif annotation of DARs was performed using HOMER v4.10.5 software[62], and the significance of enrichment was statistically determined by ZOOPS scoring (zero or one occurrence per sequence) coupled with hypergeometric enrichment calculations. GREAT analysis with single-nearest genes option was used to perform functional annotation of DARs.

**CMS-IP-seq library preparation and data analysis**. CMS-IP-seq libraries were performed as previously described with minor modifications[63]. Purified genomic DNA (with 5% mouse DNA and 0.5% lambda DNA spike-in) was sheared to 200-500 bp fragments using an M220 Focused-ultrasonicator (Covaris). Bisulfite-converted DNA libraries with methylated adapters were enriched using an in-house anti-CMS antibody bound to protein A/G Dynabeads. Amplified libraries were purified by AmpuXP beads (Beckman Coulter) and then sequenced using the Illumina NextSeq 500 instrument (75 and 40 cycles, single-end) system.

Analysis of CMS-IP-seq data was performed by in-house software 'HaMiP' v0.1.1 (https://github.com/lijinbio/HaMiP). Briefly, raw reads were mapped to hg38 and spike-in mm10 genome references using bsmap v2.89 (options: -n 1 -q 3 -r 0)[64]. After removing PCR duplicates and reads mapped to both human and spike-in mouse genome, species-specific reads were used to perform normalization for each sample according to the spike-in size factors. The whole human genome was divided into 200-bp windows and the normalized mean wigsum in each window was calculated to call the hydroxymethylation-enriched peak regions for each sample against input control. In total, 75,324 and 503 peaks were detected in WT_PP and TKO_PP samples, respectively. Differentially DHMRs between WT_PP and TKO_PP were identified (G-test; FDR < 0.05). GREAT analysis with single-nearest genes option was used to perform functional annotation of hypo-DHMRs.

CMS-IP-seq datasets of hESCs and multiple pancreatic lineage intermediates (DE, GT, and PP) were downloaded from GSE97992[20] and mapped to the hg38 genome reference using bsmap with the same parameters as in the previous analysis. Bam2wig.py was used to transform the BAM file to normalized bigwig files (option: -t 2000000000). The average 5hmC signals within each hypo-DHMR for all stages were calculated using DeepTools[65]. Compared with hESCs, increased 5hmC peaks were defined as peaks in PP cells with a ≥1.5-fold increase.

**WGBS library preparation and data analysis**. Genomic DNA was isolated using the DNeasy Blood & Tissue kit (Qiagen). Library preparation and high-throughput sequencing were conducted by BGI (Shenzhen, China). In brief, purified genomic DNA (with 1% unmethylated lambda DNA spike-in, Promega) was sheared to a fragment size of 100-700 bp (primary size 250 bp). Sheared DNA was ligated with methylated adapters (MGIEasy WGBS Adapters-16, MGI) and subjected to bisulfite conversion using the EZ DNA Methylation-gold kit (Zymo Research). Bisulfite-converted DNA was amplified with 13 PCR cycles and purified by AMPure XP beads (Beckman Coulter). All libraries were sequenced on an MGISEQ-2000 system (100 cycles, paired-end).

For data analysis, raw reads were mapped against hg38 genome reference using bsmap v2.89[64] with paired mode (options: -n 1 -q 3 -r 0), and only uniquely mapped reads were retained. More than 26.6 million CpG sites with coverage of ≥5 reads were detected in both WT_PP and TKO_PP samples, which were used for downstream analyses. BSeQC[66] and the Mcall module in MOABS[26] was applied to perform quality control and calculate the methylation ratio for each CpG site (options: --trimWGBSEndRepairPE2Seq 40). Bisulfite conversion efficiencies were estimated using spike-in unmethylated lambda phage DNA. The Mcomp module with the default parameters was used to call significant differentially methylated CpG sites (DMCs, absolute credible difference of DNA methylation ratio >20%, adjusted $p < 0.05$). DMRs were identified with a minimum of three consecutive DMCs and the maximum distance between two consecutive DMCs is 300 bp[26]. WGBS data of hESC was downloaded from the ENCODE database[27], and the average 5mC level of each hyper-DMR in hESC was also estimated. Detected hyper-DMRs (up- and down- 1 kb) were divided into 10-bp bins, and the average methylation level of each bin was calculated for hESC, WT_PP, and TKO_PP samples, independently. The output bedGraph files from Mcall included single-base resolution DNA methylation ratios, which were transformed into a bigwig file format. Motif enrichment analysis of DMRs was performed using HOMER software, functional annotation was performed using GREAT with default settings, and many plots related to WGBS data were performed using in-house software Mmint (https://github.com/lijiacd985/Mmint).

**ChIP-seq library preparation and data analysis**. ChIP-seq was performed for FOXA2 (WT_DE, WT_GT, WT_PP, TKO_DE, and TKO_PP cells), H3K4me1 (WT_PP and TKO_PP cells), and H3K27ac (WT_PP and TKO_PP cells). Chromatin immunoprecipitation was performed as previously described[25]. In brief, ~$1 \times 10^7$ cells were crosslinked for 15 min in 1% formaldehyde solution. Cells were homogenized using a Dounce homogenizer and lysed in lysis buffer (50 mM Trish HCl, pH 8.0, 5 mM EDTA, 1% SDS, and 1× protease inhibitor cocktail) for 10 min on ice. The lysate was sonicated in a Bioruptor® Plus (Diagenode) for 12-15 min (30 s on, 30 s off) to obtain 200-1000 bp DNA fragments. Between 20 and 35 μg of the resulting sheared chromatin was incubated with 10 μg goat anti-FOXA2 (R&D AF2400), 5 μg rabbit anti-H3K27ac (Active Motif 39133), or 5 μg rabbit anti-H3K4me1 (Abcam ab8895) overnight at 4 ℃ followed by incubation with Dyna-beads (Thermo Fisher Scientific) for 4 h at 4 ℃. The crosslinks were reversed, and DNA was purified. ChIP-seq libraries were prepared using the NEB Next Ultra II DNA Library Prep kit (New England Biolabs) following the manufacturer's instruction and subjected to high-throughput sequencing on an Illumina Novaseq 6000 system (150 cycles, paired-end) at Novogene (Tianjin, China).

Quality control and alignment of raw reads of ChIP-seq data were performed similarly to the ATAC-seq data analysis described above. TrimGalore (options: --quality 20 and --length 50) was used to remove the adapter. Bowtie2 with the '--very-sensitive' option was used for alignment, and only uniquely mapped reads were retained. Bam2wig.py was used to transform the BAM file to normalized bigwig files for visualization (option: -t 2000000000). For FOXA2 ChIP-seq, Macs2[67] was used to call ChIP-seq enriched peak regions with default parameters for each replicate. The BEDTools intersect was used to obtain the highly confident FOXA2 bound regions in two biological replicates for each sample. Then BEDTools merge was used to merge the confident peaks from WT and TKO samples to create the consensus FOXA2 bound regions at the DE and PP stage, independently. The reads number was counted for consensus peaks in each replicate and DESeq2 v1.28.1 was used to call significant differential FOXA2 bound regions (|fold change| ≥ 2, FDR < 0.05). Similar to ATAC-seq, motif analysis of FOXA2 bound regions was performed using HOMER software and functional annotation was performed using GREAT.

**Integration of analyses**. The number of mapped reads, mapped ratios and other statistic information for RNA-seq, ATAC-seq, WGBS, CMS-IP-seq, and ChIP-seq were listed in Supplementary Data 11. To compare DNA methylation between TKO_PP and WT_PP cells at annotated genomic features, WGBS data of H1 hESCs was downloaded from ENCODE database, ChIP-seq datasets of GATA4, GATA6, and PDX1 were downloaded from GSE117136, and ChIP-seq datasets of HNF6 was downloaded from GSE149148 (Supplementary Table 4). Bivalent promoters and poised and active enhancers in hESC-derived pancreatic progenitors were down-loaded from EMTAB1086 and GSE54471, respectively. Data were analyzed as described above. The methylation difference between WT_PP and TKO_PP across each genomic feature was performed by deepTools and Mmint software. The signal distribution across multiple different regions was performed by deepTools.

For integrated data visualization, bigwig files from RNA-seq, ATAC-seq, WGBS, CMS-IP-seq, and ChIP-seq experiments were uploaded to the University of California, Santa Cruz (UCSC) genome browser. R package ggplot2 was used to plot volcano, bar, boxplot, point, heatmap, and scatter plots[61].

**Quantification and statistical analysis**. Statistical analyses were performed using GraphPad Prism (v1.28.1). Statistical parameters such as the value of n, mean, standard deviation (SD), significance level (ns, not significant; *$p < 0.05$; **$p < 0.01$; ***$p < 0.001$, and ****$p < 0.0001$), and the statistical tests used are reported in the figures and figure legends. Unless otherwise noted, the "$n$" refers to the number of independent differentiation experiments. Statistical analyses for RNA-seq, WGBS, CMS-IP-seq, and ChIP-seq data are described in the corresponding sections.

**Reporting summary**. Further information on research design is available in the Nature Research Reporting Summary linked to this article.

## Data availability

The pertinent data supporting the findings of this study are available from the corresponding author upon reasonable request. All ChIP-seq, CMS-IP-seq, RNA-seq, ATAC-seq, and WGBS datasets generated for this study have been deposited at Gene Expression Omnibus (GEO) under the accession number GSE146486. The following datasets used in this study were obtained from GEO and ArrayExpress repositories, CMS-IP-seq: DE, GT, PP (GSE97992); WGBS: H1 hESC from ENCODE project (GSE80911); ChIP-seq: H3K4me3 in PP (E-MTAB-1086), H3K27me3 in PP (E-MTAB-1086), GATA4 in DE, GT, PP (GSE117136), GATA6 in DE, GT, PP (GSE117136), PDX1 in PP (GSE117136); HNF6 in PP (GSE149148). Source data are provided with this paper.

## Code availability

Analysis code that accompanies this paper is provided on Github (https://github.com/lijinbio/HaMiP[68] and https://github.com/lijiacd985/Mmint[69]).

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

## Acknowledgements

We thank the University of Macau, Faculty of Health Sciences, Animal Research Facility for animal housing. Mycoplasma detection was performed by the Stem Cell Core Facility at the University of Macau. This work was supported by the National Natural Science Foundation of China (NSFC 31701276; NSFC 81773012) and The Science and Technology Development Fund, Macau SAR (0022/2019/AMJ) to R.X. The authors also thank the members of the Macau Society for Stem Cell Research (MSSCR) for inspiring discussion.

## Author contributions

R.X. conceived the project. R.X. and D.-Q.S. directed and oversaw the project. X.W. and J.K. performed experiments and collected data. J.L. performed computational analysis for WGBS, RNA-seq, CMS-IP-seq, ATAC-seq, and ChIP-seq. M.L. prepared the CMS-IP library and performed sequencing. Q.L. assisted with the animal studies. J.L. provided expert advice on bioinformatic analyses. Y.H. and J.Y. critically reviewed the manuscript. R.X., J.L., and X.W. wrote the manuscript; all other authors provided editorial advice.

## Competing interests

The authors declare no competing interests.
