## [Peer Review File · Nature Communications]

TET1 dioxygenase is required for FOXA2-associated chromatin remodeling in pancreatic beta-cell differentiationREVIEWER COMMENTS

Reviewer #1 (Remarks to the Author):

In this manuscript, Wu et al. investigate the role of TET dioxygenases in pancreatic endocrine cell specification. They generated a TET1/TET2/TET3 $-/-$ (TKO) hESC line and demonstrate that loss of TET activity leads to significant reduction in the numbers of PDX1+/NKX6.1+ pancreatic progenitor cells and a reduction in hormone+ cells, indicating a requirement for TET dioxygenases in endocrine cell specification. This defect correlates with cell type-specific decreases in hydroxymethylated regions and increases in hypermethylated regions in TKO cells. These regions also show reduced chromatin accessibility and H3K27 acetylation, suggesting a loss of active chromatin regions. The same regions are enriched for binding by FOXA2. Since FOXA2 is a pioneer TF that can contact closed chromatin, the authors propose that TET dioxygenases interact with FOXA TFs during endocrine cell specification to generate regions of open chromatin. In the second part of the study, the authors demonstrate that this phenotype is mainly driven by TET1, establishing this enzyme as the dominant TET dioxygenase during pancreatic endocrine cell development.

Since this is the first study to investigate the function of TET dioxygenases in pancreas development, the work is novel and relevant to the fields of epigenomics and cell differentiation. Overall, the experiments are thoroughly conducted and provide mostly convincing evidence to support the conclusions. However, several key points require clarification or additional evidence to make the work suitable for publication. In particular, changes in 5mHc and DNA methylation need to be better integrated and related to TET1 binding. Furthermore, the analysis of T2D variant enrichment at TET-regulated sites is unconvincing.

Major comments:

1. The authors propose that the phenotype is explained by the specific interaction of TET dioxygenases with pioneer factors during pancreas induction. The argument is based on the following evidence:

a) Regions of decreased hydroxymethylation (Hypo-DHMRs) in the TKO show enrichment for FOXA2 motifs and overlap with FOXA2 binding sites.

b) Hypo-DHMRs overlapping FOXA2 peaks show a reduction in ATAC-seq, H3K4me1, and H3K27ac signal.

c) Hyper-DMRs are enriched for FOXA2, GATA4, and GATA6 binding.

d) TET1 and FOXA2 co-bound regions have lower levels of 5mC and higher levels of ATAC, H3K4me1, and H3K27ac signal compared to regions bound by FOXA2 alone.

However, none of the experiments and analyses address whether the interaction is specific to pioneer TFs. Given that GATA TFs and FOXA TFs are core components of the transcriptional complexes at enhancers during endoderm and pancreas differentiation, it is possible that TETs are simply necessary for enhancers activation rather than interacting specifically with FOXA and other pioneer TFs as proposed. The motif enrichment pattern could simply be a reflection of the active chromatin landscape at each differentiation stage. The enrichment of HNF6 motifs at Hypo-DHMRs specific to pancreatic progenitors (Figure 2a) supports this view. The authors should either substantiate the specificity of TET interactions with pioneer TFs or change their conclusions throughout the text.

2. In Figure 2d, the authors show FOXA2 signal at identified groups of hypo-DHMRs. It would be useful to integrate existing datasets for ATAC-seq, H3K4me1 and H3K27ac ChIP-seq at DE, GT, and PP stages to examine whether chromatin accessibility and enhancer activity show the same temporal patterns as FOXA2 binding in each group of hypo-DHMRs.

3. The authors examine differences in methylation between control and TET1/TET2/TET3 $-/-$ cells by assaying both 5mHc and DNA methylation, but do not investigate whether the two marks are concordantly changed. For example, what percentage of hypo-DHMRs are also hyper-DMRs? Is there a general correlation, as one would expect?

4. The authors report that 774 hyper-DMRs in islets that are enriched for T2D risk variants overlap with hyper-DMRs in TET-deficient cells. It is unclear whether this overlap constitutes a significant enrichment and how this finding furthers our understanding of T2D pathogenesis. A useful metric would be to test whether hyper-DMRs in TET-deficient cells are statistically enriched for T2D-associated hyper-DMRs in islets (i.e. by comparing overlap of T2D-associated hyper-DMRs in islets

with a randomly selected set of methylated regions that consists of the same number/size of regions as hyper-DMRs in TET-deficient progenitors). Still, this would leave unresolved when these T2D risk variants exert their function. More informative would be to test whether hyper-DMRs in TET-deficient progenitors are enriched for T2D variants independent of hyper-DMRs in islets.

5. The authors make the argument that TET1 is the dominant TET dioxygenase relevant for pancreatic development. This claim could be bolstered by examining what percentage of TET1 binding sites overlap with the identified hypo-DHMRs and hyper-DMRs in TKO cells. It should be tested whether there is a statically significant enrichment. Furthermore, the authors should conduct analysis for TET1 binding across clusters in Figure 2b similar to what is performed for FOXA2 in Figure 2d. One would predict that TET1 binding sites in PP are enriched in the PP-specific cluster. Likewise, in Figure 3c is there a difference in TET1 binding between pancreas-specific and non-pancreatic hyper-DMRs?

6. In Figure 5f, the differences between common and PP-specific TET1 binding sites are subtle and inconsistent between different marks. Therefore, the conclusions drawn from this analysis are not entirely supported by the data. For example, proximal common binding sites appear more enriched for H3K4me3 signal than proximal PP-specific binding sites, while the opposite pattern is observed for HeK27me3 signal. The same discrepancy is observed between H3K4me1 signal and H3K27ac signal at distal binding sites. To determine whether these differences in ChIP-seq signal are significant, the authors should provide box and whisker plots and calculate p-values. They should address the observed inconsistencies.

7. The authors state that the indicated enhancer in Figure 6a is co-bound by FOXA2 and TET1. However, on the displayed genome browser track, the FOXA2 signal at the relevant enhancer is dispersed and the scale (0.46) is extremely low. This may be background signal as opposed to an actual FOXA2 peak. In all displayed genome browser snap shots, it should be indicated that highlighted regions of TF binding are identified as peaks by an unbiased peak-caller.

Minor comments:

-In Figure S1e, it should be clarified whether results from TKO clone 2 or 6 are displayed in the graph.

-The argument is made that gene expression changes in TKO cells are relatively specific to the PP stage. Although the authors provide convincing evidence that several key marker genes of the PP stage are dysregulated and that marker genes of earlier stages are relatively unaffected (Figure 1d, S1e, S1f), a genome-wide analysis should be conducted to substantiate the conclusion. For example, the authors could conduct PCA analysis of control and TKO cells at each stage of differentiation.

-Figure 2b shows that many of the identified hypo-DHMRs have the strongest 5hmC signal at the DE stage in control cells. How do the authors explain that loss of hydroxymethylation at these sites does not impair DE formation? This should be discussed.

-The heatmaps in Figure 2d are redundant with the information displayed in the density plots. I recommend removing the heatmaps.

-The methods section indicates that a 2D differentiation protocol was used. However, the immunostaining in Figure 4c appears to be from 3D aggregates. Can the authors explain?

-In Figure 4a, expression of other pancreatic TFs (see Figure S1e) should be analyzed in addition to PAX4 to determine whether there is any specificity to PAX4. If other TFs are regulated, the conclusions about TET1 acting specifically by regulating PAX4 should be tempered.

-In Figure 5d, the authors compare enriched motifs in ES-specific and PP-specific TET1 binding sites. A more direct way of performing this comparison would be to identify motifs enriched in ES-specific TET1 peaks against a background of PP-specific ChIP-seq peaks and vice versa.

-line 281, "TET 1 bound more strongly"; line 282, "Higher active chromatin signals" - Language is used that infers quantitative changes, but no quantitative analysis is performed (see comment 6).

-line 86, "ARX was not affected" - Figure S1e shows upregulation

-line 241, "significantly inhibited" - downregulated

Reviewer #2 (Remarks to the Author):

In the present study, Wu and co-authors explore the function of TET enzymes in pancreatic specification by using a human embryonic stem cell differentiation system. They find that the loss of all three TET family members significantly impair the differentiation of pancreatic beta cells. By multiple omics analysis they identify cell type-specific hypermethylated regions with altered chromatin features. Furthermore, transduction of TET1 in TET-deficient cells effectively rescued beta cell differentiation and prevented hypermethylation. ChIP-seq mapping of TET1 showed that TET1 co-localized on a subset of FOXA2 targets featuring high levels of active chromatin modifications in pancreatic progenitors.

The manuscript is clearly written and most of the conclusions are supported by reasonable experimental evidence. Although the link between transcription factor (TF) binding and DNA hydroxymethylation by TET enzymes was already well established in other publications, the question whether TET enzymes are needed for TF-mediated chromatin remodeling is less clear. Unfortunately, the authors fail to provide enough experimental evidence to support this claim. Thus, the following points need to be addressed to convincingly demonstrate that TET1 is required for FOXA2-associated chromatin remodeling in pancreatic beta cell differentiation.

- 1) The authors use a previously generated FOXA2 ChIP-seq profile (Lee et al. Cell Rep, 2019), which was obtained in a similar but not identical differentiation system. I see the precise mapping of FOXA2 binding sites key for elucidating the link with TET1. As FOXA2 binding is highly cell type-specific, slight differences in differentiation conditions can lead to redistribution of FOXA2 to other sites. It is also very important to consider that TET tko cell display strong differentiation defects, which are likely to result in altered FOXA2 binding and, in turn, altered chromatin accessibility. Therefore, I see it crucial to obtain FOXA2 ChIP-seq data in wild type and TET ko situations.
- 2) Since TET ko impairs pancreatic endoderm differentiation it is difficult to distinguish between direct and indirect effects on FOXA2 binding sites. As mentioned in (1), FOXA2 could be redistributed to other binding sites, leading to reduced accessibility on original peaks. Other effects may be in place here. For example, in different experimental systems it was shown that co-binding of transcription factors is needed to establish chromatin accessibility. In the human beta cell differentiation system, FOXA2 is likely to pair with beta cell specific transcription factors. Lack of those TFs or redistribution of them may also result in reduced chromatin accessibility on FOXA2 binding sites. To better disentangle the interplay between TET and FOXA2 it would be necessary to investigate an experimental system with minimal phenotypic and transcriptional changes upon TET ko. For example, the authors may investigate FOXA2 binding and chromatin accessibility in wildtype vs TET ko human ES cells.
- 3) ChIP-seq for TET1 resulted in only partial overlap with FOXA2 binding sites. It would be important to show quality controls for the specificity of the TET1 antibody.

Minor points:

- 4) lines 165-166 "suggesting that inhibition of TET did not alter the expression levels of pioneer TFs". Only data for FOXA and GATA are shown.
- 5) lanes 181-182: "consistent with a suggested connection between hypomethylation and activation of transposable elements". Are transposable elements upregulated in TET KO cells?
- 6) lines 192-193: "we analyzed differentially methylated regions (DMRs) by connecting at least three consecutive DMCs". What is the rationale for this approach?
- 7) Fig 3d,e: This density plots should be accompanied by heat maps showing the signal distribution across the considered genomic regions.
- 8) line 200: "demethylation is primarily associated with pioneer TF binding". Did the author check for the presence of non-pioneering TFs? Here, it would be nice to decipher co-binding of TFs (related to point (2)).
- 9) Supplementary Fig 3h and Fig 6a: The signal scale for FOXA2 track is set to 0.87 and 0.59 respectively while in Fig 3g is set to 88. In Fig 3g the Foxa2 signal appears like a robust peak, but in the other figures, the Foxa2 track basically shows background signal (which would become obvious with using the same scale). The claim in lines 874-876 that Fig. 6a represents a FOXA2/TET1 co-bound region needs to be reconsidered.
- 10) Supplementary Fig 4a: one can hardly appreciate differences between these pictures. A quantification of the signals could be provided.
- 11) Fig 4f: how many mice have been used for this experiment? Which statistics? Number of replicates, statistics and p-values should be provided in each figure legend.

12) lines 382-83: "FOXA2 can recruit nucleosome remodeling complexes SWI/SNF to alter the surrounding chromatin structure (ref.45)". This reference is wrong and needs to be corrected.

13) ChIP-seq protocol described in the methods would need more details.

14) lines 594-596: "Integration of analyses. RNA-seq, WGBS, CMS-IP-seq, ATAC-seq and ChIP-seq library preparations were performed as previously described. Detail experimental procedures can be found in Supplementary Materials". The reference is lacking and I could not find additional details of the experimental procedures in the supplements.

General Response to Reviewers:

We thank all reviewers for their positive and helpful comments. We were enthused by the reviewers' generally positive response to our work. We concur with the reviewers that the study would benefit from further insight into the interaction of TET1 with different TFs and FOXA2 occupancy in wildtype and TET-knockout cells. Since first submission of the manuscript, we conducted genome-wide analysis of FOXA2 occupancy and co-immunoprecipitation of TET1. Integrative analysis of FOXA2 distribution with chromatin state in wildtype and TET-depleted cells at the definitive endoderm and pancreatic progenitor stages has revealed informative mechanistic insight into how FOXA2 and TETs cooperatively control cell lineage decisions.

Key novel findings that have emerged from our analysis include:

1. We mapped FOXA2 occupancy by ChIP-seq in WT and TKO cells at the DE and PP stages. We found that FOXA2 binding was dramatically changed at the PP stage but not at the DE stage in TET-deficient cells. We discover that 1) de novo FOXA2 recruitment at genomic loci with low levels of active chromatin modifications is decreased upon TET depletion, and 2) FOXA2-decreased binding sites are enriched with bHLH TFs, such as NEUROD1 and PTF1A, which fail to be induced in TET-knockout cells.
2. We identify specific interaction between TET1 and FOXA2 by co-immunoprecipitation in endodermal lineage intermediates, while co-precipitation of other endoderm proteins, such as SOX17, GATA6, and FOXA1, is not observed.
3. We show that transduction of full-length *TET1* but not the TET1-catalytic-domain in TET-depleted cells effectively rescues β -cell differentiation accompanied by restoring *PAX4* hypomethylation and expression.

Regarding the specificity of TET1 ChIP-seq, we were not able to verify it because the TET1 antibody (Sigma 09-872) we used for ChIP-seq was discontinued. We tried to examine other commercially available TET1 antibodies, but none of them shown specific pull-down in WT cells when compared to TET1-knockout cells. To ensure the best possible accuracy for TET1 occupancy, we decided to not include the TET1 ChIP-seq results into the revised manuscript and substantially revised our manuscript with new experimental results. Below we provide point-by-point responses to the concerns raised by each reviewer, and actions taken to address each concern.

Reviewer #1 (Remarks to the Author):

In this manuscript, Wu et al. investigate the role of TET dioxygenases in pancreatic endocrine cell specification. They generated a TET1/TET2/TET3 *-/-* (TKO) hESC line and demonstrate that loss of TET activity leads to significant reduction in the numbers of PDX1+/NKX6.1+ pancreatic progenitor cells and a reduction in hormone+ cells, indicating a requirement for TET dioxygenases in endocrine

cell specification. This defect correlates with cell type-specific decreases in hydroxymethylated regions and increases in hypermethylated regions in TKO cells. These regions also show reduced chromatin accessibility and H3K27 acetylation, suggesting a loss of active chromatin regions. The same regions are enriched for binding by FOXA2. Since FOXA2 is a pioneer TF that can contact closed chromatin, the authors propose that TET dioxygenases interact with FOXA TFs during endocrine cell specification to generate regions of open chromatin. In the second part of the study, the authors demonstrate that this phenotype is mainly driven by TET1, establishing this enzyme as the dominant TET dioxygenase during pancreatic endocrine cell development.

Since this is the first study to investigate the function of TET dioxygenases in pancreas development, the work is novel and relevant to the fields of epigenomics and cell differentiation. Overall, the experiments are thoroughly conducted and provide mostly convincing evidence to support the conclusions. However, several key points require clarification or additional evidence to make the work suitable for publication. In particular, changes in 5mC and DNA methylation need to be better integrated and related to TET1 binding. Furthermore, the analysis of T2D variant enrichment at TET-regulated sites is unconvincing.

Major comments:

1. The authors propose that the phenotype is explained by the specific interaction of TET dioxygenases with pioneer factors during pancreas induction. The argument is based on the following evidence:

a) Regions of decreased hydroxymethylation (Hypo-DHMRs) in the TKO show enrichment for FOXA2 motifs and overlap with FOXA2 binding sites.

b) Hypo-DHMRs overlapping FOXA2 peaks show a reduction in ATAC-seq, H3K4me1, and H3K27ac signal.

c) Hyper-DMRs are enriched for FOXA2, GATA4, and GATA6 binding.

d) TET1 and FOXA2 co-bound regions have lower levels of 5mC and higher levels of ATAC, H3K4me1, and H3K27ac signal compared to regions bound by FOXA2 alone.

However, none of the experiments and analyses address whether the interaction is specific to pioneer TFs. Given that GATA TFs and FOXA TFs are core components of the transcriptional complexes at enhancers during endoderm and pancreas differentiation, it is possible that TETs are simply necessary for enhancers activation rather than interacting specifically with FOXA and other pioneer TFs as proposed. The motif enrichment pattern could simply be a reflection of the active chromatin landscape at each differentiation stage. The enrichment of HNF6 motifs at Hypo-DHMRs specific to pancreatic progenitors (Figure 2a) supports this view. The authors should either substantiate the specificity of TET interactions with pioneer TFs or change their conclusions throughout the text.

2. In Figure 2d, the authors show FOXA2 signal at identified groups of hypo-DHMRs. It would be useful to integrate existing datasets for ATAC-seq, H3K4me1 and H3K27ac ChIP-seq at DE, GT, and

PP stages to examine whether chromatin accessibility and enhancer activity show the same temporal patterns as FOXA2 binding in each group of hypo-DHMRs.

3. The authors examine differences in methylation between control and TET1/TET2/TET3 $-/-$ cells by assaying both 5mC and DNA methylation, but do not investigate whether the two marks are concordantly changed. For example, what percentage of hypo-DHMRs are also hyper-DMRs? Is there a general correlation, as one would expect?

4. The authors report that 774 hyper-DMRs in islets that are enriched for T2D risk variants overlap with hyper-DMRs in TET-deficient cells. It is unclear whether this overlap constitutes a significant enrichment and how this finding furthers our understanding of T2D pathogenesis. A useful metric would be to test whether hyper-DMRs in TET-deficient cells are statistically enriched for T2D-associated hyper-DMRs in islets (i.e. by comparing overlap of T2D-associated hyper-DMRs in islets with a randomly selected set of methylated regions that consists of the same number/size of regions as hyper-DMRs in TET-deficient progenitors). Still, this would leave unresolved when these T2D risk variants exert their function. More informative would be to test whether hyper-DMRs in TET-deficient progenitors are enriched for T2D variants independent of hyper-DMRs in islets.

5. The authors make the argument that TET1 is the dominant TET dioxygenase relevant for pancreatic development. This claim could be bolstered by examining what percentage of TET1 binding sites overlap with the identified hypo-DHMRs and hyper-DMRs in TKO cells. It should be tested whether there is a statistically significant enrichment. Furthermore, the authors should conduct analysis for TET1 binding across clusters in Figure 2b similar to what is performed for FOXA2 in Figure 2d. One would predict that TET1 binding sites in PP are enriched in the PP-specific cluster. Likewise, in Figure 3c is there a difference in TET1 binding between pancreas-specific and non-pancreatic hyper-DMRs?

6. In Figure 5f, the differences between common and PP-specific TET1 binding sites are subtle and inconsistent between different marks. Therefore, the conclusions drawn from this analysis are not entirely supported by the data. For example, proximal common binding sites appear more enriched for H3K4me3 signal than proximal PP-specific binding sites, while the opposite pattern is observed for H3K27me3 signal. The same discrepancy is observed between H3K4me1 signal and H3K27ac signal at distal binding sites. To determine whether these differences in ChIP-seq signal are significant, the authors should provide box and whisker plots and calculate p-values. They should address the observed inconsistencies.

7. The authors state that the indicated enhancer in Figure 6a is co-bound by FOXA2 and TET1. However, on the displayed genome browser track, the FOXA2 signal at the relevant enhancer is dispersed and the scale (0.46) is extremely low. This may be background signal as opposed to an actual FOXA2 peak. In all displayed genome browser snapshots, it should be indicated that highlighted regions of TF binding are identified as peaks by an unbiased peak-caller.

Minor comments:

- In Figure S1e, it should be clarified whether results from TKO clone 2 or 6 are displayed in the graph.
- The argument is made that gene expression changes in TKO cells are relatively specific to the PP stage. Although the authors provide convincing evidence that several key marker genes of the PP stage are dysregulated and that marker genes of earlier stages are relatively unaffected (Figure 1d, S1e, S1f), a genome-wide analysis should be conducted to substantiate the conclusion. For example, the authors could conduct PCA analysis of control and TKO cells at each stage of differentiation.
- Figure 2b shows that many of the identified hypo-DHMRs have the strongest 5hmC signal at the DE stage in control cells. How do the authors explain that loss of hydroxymethylation at these sites does not impair DE formation? This should be discussed.
- The heatmaps in Figure 2d are redundant with the information displayed in the density plots. I recommend removing the heatmaps.
- The methods section indicates that a 2D differentiation protocol was used. However, the immunostaining in Figure 4c appears to be from 3D aggregates. Can the authors explain?
- In Figure 4a, expression of other pancreatic TFs (see Figure S1e) should be analyzed in addition to PAX4 to determine whether there is any specificity to PAX4. If other TFs are regulated, the conclusions about TET1 acting specifically by regulating PAX4 should be tempered.
- In Figure 5d, the authors compare enriched motifs in ES-specific and PP-specific TET1 binding sites. A more direct way of performing this comparison would be to identify motifs enriched in ES-specific TET1 peaks against a background of PP-specific ChIP-seq peaks and vice versa.
- line 281, “TET 1 bound more strongly”; line 282, “Higher active chromatin signals” - Language is used that infers quantitative changes, but no quantitative analysis is performed (see comment 6).
- line 86, “ARX was not affected” – Figure S1e shows upregulation
- line 241, “significantly inhibited” - downregulated

Response to Reviewer #1

Major comments:

1. The authors propose that the phenotype is explained by the specific interaction of TET dioxygenases with pioneer factors during pancreas induction. The argument is based on the following evidence:
 - a) Regions of decreased hydroxymethylation (Hypo-DHMRs) in the TKO show enrichment for FOXA2 motifs and overlap with FOXA2 binding sites.
 - b) Hypo-DHMRs overlapping FOXA2 peaks show a reduction in ATAC-seq, H3K4me1, and H3K27ac signal.
 - c) Hyper-DMRs are enriched for FOXA2, GATA4, and GATA6 binding.
 - d) TET1 and FOXA2 co-bound regions have lower levels of 5mC and higher levels of ATAC,

H3K4me1, and H3K27ac signal compared to regions bound by FOXA2 alone.

However, none of the experiments and analyses address whether the interaction is specific to pioneer TFs. Given that GATA TFs and FOXA TFs are core components of the transcriptional complexes at enhancers during endoderm and pancreas differentiation, it is possible that TETs are simply necessary for enhancers activation rather than interacting specifically with FOXA and other pioneer TFs as proposed. The motif enrichment pattern could simply be a reflection of the active chromatin landscape at each differentiation stage. The enrichment of HNF6 motifs at Hypo-DHMRs specific to pancreatic progenitors (Figure 2a) supports this view. The authors should either substantiate the specificity of TET interactions with pioneer TFs or change their conclusions throughout the text.

Response: We agree with reviewer that additional analysis on co-binding of TET and TFs will further strengthen the conclusion. To determine potential interaction between TET1 and FOXA2, we performed co-immunoprecipitation experiment. Due to the lack of proper TET1 antibody for IP, we applied an anti-FLAG antibody to pull down TET1 in the TKO-TET1FL cells which expressed a FLAG-tagged full-length TET1 gene. We have demonstrated that transduction of full-length *TET1* in TET-deficient cells effectively rescues β -cell differentiation accompanied by restoring *PAX4* hypomethylation and expression (**Fig. 6; Supplementary Fig. 6**). It is therefore reasonable, although not ideal, to apply a FLAG antibody to pull down TET1 and examine its interaction partners in TKO-TET1FL cells. All tested endodermal TFs, including FOXA2, FOXA1, GATA6, and SOX17, were expressed at similar levels between TKO-TET1FL and TKO cells at the DE stage. Of particular interest was the observation that only FOXA2 but not FOXA1, GATA6, and SOX17 was co-precipitated with TET1 suggesting TET1 specifically interacts with FOXA2 in endodermal lineage intermediates. We show the new analysis of TET1 coimmunoprecipitation in **Fig. 6c** and **Supplementary Fig. 6d** of the revised manuscript.

2. In Figure 2d, the authors show FOXA2 signal at identified groups of hypo-DHMRs. It would be useful to integrate existing datasets for ATAC-seq, H3K4me1 and H3K27ac ChIP-seq at DE, GT, and PP stages to examine whether chromatin accessibility and enhancer activity show the same temporal patterns as FOXA2 binding in each group of hypo-DHMRs.

Response: As suggested by the reviewer, we integrated existing datasets for ATAC-seq (GSE114101), H3K4me1 ChIP-seq (GSE117136), and H3K27ac ChIP-seq (GSE117136) and analyzed their temporal patterns at ‘differentiation-specific hypo-DHMRs’ clusters identified in **Fig. 3b**. Consistent with correspondence between the dynamic distribution of FOXA2 and 5hmC (**Fig. 3d**), we found that ATAC-seq, H3K4me1, and H3K27ac signals showed concordant changes with 5hmC signals in most groups, but not as precise as the temporal patterns of FOXA2 (**Figure 1; to reviewer only**).

Figure 1. Average density plots showing dynamic changes of ATAC-seq (GSE114101), H3K4me1 (GSE117136), and H3K27ac (GSE117136) signals at the DE, GT, and PP stages in each cluster defined in Fig. 3b.

3. The authors examine differences in methylation between control and TET1/TET2/TET3 ^{-/-} cells by assaying both 5mC and DNA methylation, but do not investigate whether the two marks are concordantly changed. For example, what percentage of hypo-DHMRs are also hyper-DMRs? Is there a general correlation, as one would expect?

Response: We appreciate the reviewer's constructive comment. We analyzed changes of DNA methylation in hypo-hydroxymethylated regions and found 93% of hypo-DHMRs displayed increased DNA methylation in TET-deficient cells (Figure 2a; to reviewer only). Consistently, 90% of hyper-DMRs showed decreased 5hmC levels in TKO_PP cells (Figure 2b; to reviewer only). Taken together, these data suggest that hyper-methylation and hypo-hydroxymethylation is general correlated.

a

b

Figure

2. a

Scatterplots presenting mean methylation ratio (5mC/C) in WT_PP cells versus TKO_PP cells at hypo-DHMRs identified in TKO_PP cells. Percentage of regions with increased methylation is indicated

(methylation ratio difference > 0). **b** Scatterplot showing log₂ of 5hmC signals in WT versus TKO at hyper-DMRs identified in TKO_PP cells. Red indicates increased 5hmC and blue indicates decreased 5hmC in TKO_PP relative to WT_PP (fold changel ≥ 2).

4. The authors report that 774 hyper-DMRs in islets that are enriched for T2D risk variants overlap with hyper-DMRs in TET-deficient cells. It is unclear whether this overlap constitutes a significant enrichment and how this finding furthers our understanding of T2D pathogenesis. A useful metric would be to test whether hyper-DMRs in TET-deficient cells are statistically enriched for T2D-associated hyper-DMRs in islets (i.e. by comparing overlap of T2D-associated hyper-DMRs in islets with a randomly selected set of methylated regions that consists of the same number/size of regions as hyper-DMRs in TET-deficient progenitors). Still, this would leave unresolved when these T2D risk variants exert their function. More informative would be to test whether hyper-DMRs in TET-deficient progenitors are enriched for T2D variants independent of hyper-DMRs in islets.

Response: As suggested, we perform systematic analysis of hyper-DMRs in TKO_PP versus T2D-associated hyper-DMRs in islets. We found that although hyper-DMRs in TKO_PP were significantly enriched for T2D-associated hyper-DMRs, less than 6% of the regions were overlapped (**Figure 3; for reviewer only**). It is therefore difficult to conclude whether those overlapped hyper-DMRs exert their function at developmental stages. The reviewer also suggests testing whether hyper-DMRs in TET-deficient pancreatic progenitors contributes to T2D pathogenesis. We total agree with the reviewer, yet our current study mainly focuses on the epigenetic regulation of TETs in pancreatic endocrine lineage specification. We decided to not include the contexts related to T2D in the revised manuscript and will conduct more comprehensive analysis in our following studies.

Figure 3. Venn diagrams depicting hyper-DMRs in TET-deficient pancreatic progenitors overlapped with T2D-associated hyper-DMRs (left) or shuffled regions (right), demonstrating hyper-DMRs in TKO_PP are significant enriched for T2D-associated hyper-DMRs. The shuffled regions were generated by Bedtools shuffle to obtain randomly distributed regions of T2D-associated hyper-DMRs across whole genome. Bedtools intersect was used to define overlapped regions between hyper-DMRs in TKO_PP cells and shuffled regions.

5. The authors make the argument that TET1 is the dominant TET dioxygenase relevant for pancreatic development. This claim could be bolstered by examining what percentage of TET1 binding sites overlap with the identified hypo-DHMRs and hyper-DMRs in TKO cells. It should be tested whether there is a statically significant enrichment. Furthermore, the authors should conduct analysis for TET1 binding across clusters in Figure 2b similar to what is performed for FOXA2 in Figure 2d. One would predict that TET1 binding sites in PP are enriched in the PP-specific cluster. Likewise, in Figure 3c is there a difference in TET1 binding between pancreas-specific and non-pancreatic hyper-DMRs?

Response: Because the antibody (Sigma 09-872) we used for TET1-ChIP seq was discontinued, we were not able to perform additional experiment to verify its specificity. To ensure the best possible accuracy for TET1 occupancy, we decided to not include the TET1 ChIP-seq results in the revised manuscript.

6. In Figure 5f, the differences between common and PP-specific TET1 binding sites are subtle and inconsistent between different marks. Therefore, the conclusions drawn from this analysis are not entirely supported by the data. For example, proximal common binding sites appear more enriched for H3K4me3 signal than proximal PP-specific binding sites, while the opposite pattern is observed for H3K27me3 signal. The same discrepancy is observed between H3K4me1 signal and H3K27ac signal at distal binding sites. To determine whether these differences in ChIP-seq signal are significant, the authors should provide box and whisker plots and calculate p-values. They should address the observed inconsistencies.

Response: To ensure the best possible accuracy for TET1 occupancy, we decided to not include the TET1 ChIP-seq results in the revised manuscript.

7. The authors state that the indicated enhancer in Figure 6a is co-bound by FOXA2 and TET1. However, on the displayed genome browser track, the FOXA2 signal at the relevant enhancer is dispersed and the scale (0.46) is extremely low. This may be background signal as opposed to an actual FOXA2 peak. In all displayed genome browser snap shots, it should be indicated that highlighted regions of TF binding are identified as peaks by an unbiased peak-caller.

Response: We thank reviewer for pointing out these issues. We implied our own FOXA2 ChIP-seq data to replace the previously generated FOXA2 ChIP-seq data (GSE117136) ¹. All highlighted regions of FOXA2 binding in genome-browser views of different loci were identified as peaks by MACS2 (**Fig. 2g, Supplementary Fig. 2h, Fig. 6d**).

Minor comments:

-In Figure S1e, it should be clarified whether results from TKO clone 2 or 6 are displayed in the graph.

Response: These results were generated from TKO clone 2. As suggested by the reviewer, we clarified it in the figure legend of **Supplementary Fig.1e**.

-The argument is made that gene expression changes in TKO cells are relatively specific to the PP stage. Although the authors provide convincing evidence that several key marker genes of the PP stage are dysregulated and that marker genes of earlier stages are relatively unaffected (Figure 1d, S1e, S1f), a genome-wide analysis should be conducted to substantiate the conclusion. For example, the authors could conduct PCA analysis of control and TKO cells at each stage of differentiation.

Response: We greatly appreciate the reviewer for suggesting these experiments. We have added results of principal component analysis in **Fig. 1d**. We also generated RNA-seq data sets from WT and TKO cells differentiated at the GT stages and used PCA to determine which stages best distinguish between these cell populations. We believe that these results strengthen our conclusions, as they suggest that TET-deficient cells show similar transcriptome at the ES, DE, and GT stages but differed substantially at the PP stage.

-Figure 2b shows that many of the identified hypo-DHMRs have the strongest 5hmC signal at the DE stage in control cells. How do the authors explain that loss of hydroxymethylation at these sites does not impair DE formation? This should be discussed.

Response: We are grateful for Reviewer 1's suggestions and have elaborated this point in discussion section (line 394-400) with the statement to now read:

“Even 5hmC was also dynamically enriched at the DE stage (**Fig. 3b**), we did not find that loss of TETs impaired definitive endoderm formation. In line with our findings, previous studies have revealed that depletion of TETs in mouse ESCs inhibits adoption of neural cell fate while concomitantly skews toward mesoderm lineage ^{2,3}. Indeed, it has been recently demonstrated that Nodal signaling, which promotes endoderm and mesoderm differentiation ^{4,5}, is hyperactivated in TET-deficient mouse embryos ⁶ suggesting activation of Nodal signaling might sustain endoderm formation upon TET depletion. “

-The heatmaps in Figure 2d are redundant with the information displayed in the density plots. I recommend removing the heatmaps.

Response: We agree with Reviewer 1, while Reviewer 2 asked for heat maps to show signal distribution across the considered genomic regions. We have kept both the density plots and heat maps in the revised manuscript.

-The methods section indicates that a 2D differentiation protocol was used. However, the immunostaining in Figure 4c appears to be from 3D aggregates. Can the authors explain?

Response: We apologize for omissions of the 3D differentiation in method section. We mainly used 2D differentiation strategy in this study. To perform immunofluorescence staining and FACS analysis on the same differentiated cells, we also applied 3D differentiation protocol. We have amended the corresponding method sections and included the 3D differentiation protocol (line 452-457).

-In Figure 4a, expression of other pancreatic TFs (see Figure S1e) should be analyzed in addition to PAX4 to determine whether there is any specificity to PAX4. If other TFs are regulated, the conclusions about TET1 acting specifically by regulating PAX4 should be tempered.

Response: As suggested by the reviewer, we performed FACS analysis of PDX1 and NKX6.1 and qPCR analysis of *SOX9*, *PTF1A*, *FOXA2*, and *ARX*. The results of FACS and qPCR were shown in **Supplementary Fig. 5a, b, c** of the revised manuscript. We did not observe a significant decrease in PDX1, *PTF1A*, *SOX9*, *NKX6.1*, or *ARX* in TET1-knockout cells, which is consistent with our RNA-seq analysis of the TET1KO cells. The fact that a clear reduction of *PTF1A* and *SOX9* was found in TET1/2DKO cells and TET1/3 double deletion resulted in less *NKX6.1* and *PTF1A* expression suggests a compensation mechanism of TETs for the regulation of pancreatic progenitor markers, such as *PTF1A*, *NKX6.1*, and *SOX9*.

-In Figure 5d, the authors compare enriched motifs in ES-specific and PP-specific TET1 binding sites. A more direct way of performing this comparison would be to identify motifs enriched in ES-specific TET1 peaks against a background of PP-specific ChIP-seq peaks and vice versa.

Response: To ensure the best possible accuracy for TET1 occupancy, we decided to not include the TET1 ChIP-seq results in the revised manuscript.

-line 281, “TET 1 bound more strongly”; line 282, “Higher active chromatin signals” - Language is used that infers quantitative changes, but no quantitative analysis is performed (see comment 6).

Response: To ensure the best possible accuracy for TET1 occupancy, we decided to not include the TET1 ChIP-seq results in the revised manuscript.

-line 86, “ARX was not affected” – Figure S1e shows upregulation

Response: We thank reviewer for pointing out these issues. Indeed, it is inaccurate to use “not affected”. We changed to “not inhibited” (line 93).

-line 241, “significantly inhibited” - downregulated

Response: We have corrected them in the main text (line 303).

Reviewer #2 (Remarks to the Author):

In the present study, Wu and co-authors explore the function of TET enzymes in pancreatic specification by using a human embryonic stem cell differentiation system. They find that the loss of all three TET family members significantly impair the differentiation of pancreatic beta cells. By multiple omics analysis they identify cell type-specific hypermethylated regions with altered chromatin features. Furthermore, transduction of TET1 in TET-deficient cells effectively rescued beta cell differentiation and prevented hypermethylation. ChIP-seq mapping of TET1 showed that TET1 co-localized on a subset of FOXA2 targets featuring high levels of active chromatin modifications in pancreatic progenitors.

The manuscript is clearly written and most of the conclusions are supported by reasonable experimental evidence. Although the link between transcription factor (TF) binding and DNA hydroxymethylation by TET enzymes was already well established in other publications, the question whether TET enzymes are needed for TF-mediated chromatin remodeling is less clear. Unfortunately, the authors fail to provide enough experimental evidence to support this claim. Thus, the following points need to be addressed to convincingly demonstrate that TET1 is required for FOXA2-associated chromatin remodeling in pancreatic beta cell differentiation.

1) The authors use a previously generated FOXA2 ChIP-seq profile (Lee et al. Cell Rep, 2019), which was obtained in a similar but not identical differentiation system. I see the precise mapping of FOXA2 binding sites key for elucidating the link with TET1. As FOXA2 binding is highly cell type-specific, slight differences in differentiation conditions can lead to redistribution of FOXA2 to other sites. It is also very important to consider that TET tko cell display strong differentiation defects, which are likely to result in altered FOXA2 binding and, in turn, altered chromatin accessibility. Therefore, I see it crucial to obtain FOXA2 ChIP-seq data in wild type and TET ko situations.

2) Since TET ko impairs pancreatic endoderm differentiation it is difficult to distinguish between direct and indirect effects on FOXA2 binding sites. As mentioned in (1), FOXA2 could be redistributed to other binding sites, leading to reduced accessibility on original peaks. Other effects may be in place here. For example, in different experimental systems it was shown that co-binding of transcription factors is needed to establish chromatin accessibility. In the human beta cell differentiation system, FOXA2 is likely to pair with beta cell specific transcription factors. Lack of those TFs or redistribution of them may also result in reduced chromatin accessibility on FOXA2 binding sites. To better disentangle the interplay between TET and FOXA2 it would be necessary to investigate an experimental system with minimal phenotypic and transcriptional changes upon TET ko. For example, the authors may investigate FOXA2 binding and chromatin accessibility in wildtype

vs TET ko human ES cells.

3) ChIP-seq for TET1 resulted in only partial overlap with FOXA2 binding sites. It would be important to show quality controls for the specificity of the TET1 antibody.

Minor points:

4) lines 165-166 “suggesting that inhibition of TET did not alter the expression levels of pioneer TFs”. Only data for FOXA and GATA are shown.

5) lanes 181-182: “consistent with a suggested connection between hypomethylation and activation of transposable elements”. Are transposable elements upregulated in TET KO cells?

6) lines 192-193: “we analyzed differentially methylated regions (DMRs) by connecting at least three consecutive DMCs”. What is the rationale for this approach?

7) Fig 3d,e: This density plots should be accompanied by heat maps showing the signal distribution across the considered genomic regions.

8) line 200: “demethylation is primarily associated with pioneer TF binding”. Did the author check for the presence of non-pioneering TFs? Here, it would be nice to decipher co-binding of TFs (related to point (2)).

9) Supplementary Fig 3h and Fig 6a: The signal scale for FOXA2 track is set to 0.87 and 0.59 respectively while in Fig 3g is set to 88. In Fig 3g the Foxa2 signal appears like a robust peak, but in the other figures, the Foxa2 track basically shows background signal (which would become obvious with using the same scale). The claim in lines 874-876 that Fig. 6a represents a FOXA2/TET1 co-bound region needs to be reconsidered.

10) Supplementary Fig 4a: one can hardly appreciate differences between these pictures. A quantification of the signals could be provided.

11) Fig 4f: how many mice have been used for this experiment? Which statistics? Number of replicates, statistics and p-values should be provided in each figure legend.

12) lines 382-83: “FOXA2 can recruit nucleosome remodeling complexes SWI/SNF to alter the surrounding chromatin structure (ref.45)”. This reference is wrong and needs to be corrected.

13) ChIP-seq protocol described in the methods would need more details.

14) lines 594-596: “Integration of analyses. RNA-seq, WGBS, CMS-IP-seq, ATAC-seq and ChIP-seq library preparations were performed as previously described. Detail experimental procedures can be found in Supplementary Materials”. The reference is lacking and I could not find additional details of the experimental procedures in the supplements.

Response to Reviewer #2

1) The authors use a previously generated FOXA2 ChIP-seq profile (Lee et al. Cell Rep, 2019), which was obtained in a similar but not identical differentiation system. I see the precise mapping of FOXA2 binding sites key for elucidating the link with TET1. As FOXA2 binding is highly cell type-specific,

slight differences in differentiation conditions can lead to redistribution of FOXA2 to other sites. It is also very important to consider that TET ko cell display strong differentiation defects, which are likely to result in altered FOXA2 binding and, in turn, altered chromatin accessibility. Therefore, it is crucial to obtain FOXA2 ChIP-seq data in wild type and TET ko situations.

Response: We greatly appreciate the reviewer for suggesting these experiments. We performed FOXA2 ChIP-seq in both WT and TKO cells and added the new analysis of FOXA2 occupancy in **Fig. 1i**, **Fig. 3d**, **Fig. 4**, and **Supplementary Fig. 4** of the revised manuscript. Consistent with previous reports^{1,7}, FOXA2 was primarily located at non-promotor regions (**Supplementary Fig. 4a**) implicating that FOXA2 is mainly involved in transcription regulation through distal regulatory elements. FOXA2 binding at each stage of differentiation in WT cells showed similar distribution as the previously published FOXA2 ChIP-seq results (revised **Fig. 3d** versus original **Fig. 2d**).

2) Since TET ko impairs pancreatic endoderm differentiation it is difficult to distinguish between direct and indirect effects on FOXA2 binding sites. As mentioned in (1), FOXA2 could be redistributed to other binding sites, leading to reduced accessibility on original peaks. Other effects may be in place here. For example, in different experimental systems it was shown that co-binding of transcription factors is needed to establish chromatin accessibility. In the human beta cell differentiation system, FOXA2 is likely to pair with beta cell specific transcription factors. Lack of those TFs or redistribution of them may also result in reduced chromatin accessibility on FOXA2 binding sites. To better disentangle the interplay between TET and FOXA2 it would be necessary to investigate an experimental system with minimal phenotypic and transcriptional changes upon TET ko. For example, the authors may investigate FOXA2 binding and chromatin accessibility in wildtype vs TET ko human ES cells.

Response: This is an excellent point, and as suggested we have examined FOXA2 binding in wildtype and TET-knockout cells. Because FOXA2 doesn't not express in hESCs, we investigated changes of FOXA2 binding in TKO_DE cells, which showed minimal phenotypic and transcriptional changes upon TET depletion, and TKO_PP cells which showed impaired pancreatic cell fate. The results are now

presented in **Fig. 4** and **Supplementary Fig. 4**.

We found that upon TET depletion, 99.5% FOXA2 binding sites did not show changes at the DE stage (**Fig. 4a**), while FOXA2 occupancy was dramatically changed at the PP stage, in which 10% and 6% of FOXA2 target sites showed reduced and greater FOXA2 binding, respectively (**Fig. 4a, Supplementary Fig. 4b**). Since DNA methylation and hydroxymethylation status were significantly altered in TET-deficient cells (**Fig. 2a, Supplementary Fig. 3c**), we wondered whether increases in methylation or decreases in hydroxymethylation contributed to differential recruitment of FOXA2 in TKO_PP cells. We first assessed changes in FOXA2 binding in WT_PP versus TKO_PP cells at the hyper-DMRs or hypo-DHMRs identified in TKO_PP cells. We found that ~75% regions showed no changes in FOXA2 binding (**Fig. 4b**). We then tested if only hypermethylated or hypo-hydroxymethylated FOXA2-bound sites would be affected. We found that similar proportions of FOXA2 target sites displayed reduced/greater of FOXA2 binding regardless of the presence or absence of hypermethylation (**Fig. 4c**). Similarly, hypo-hydroxymethylated FOXA2-bound sites did not show a preference in gain/loss of FOXA2 binding compared to iso-hydroxymethylated FOXA2-bound sites (**Fig. 4d**). Consistent with the ability of pioneer TFs to engage with inaccessible chromatin, our results strongly demonstrate that DNA methylation/hydroxymethylation states do not interfere in FOXA2 binding.

To comprehensively characterize temporal patterns of FOXA2 recruitment at the FOXA2-decreased binding sites, we clustered FOXA2 binding signal in WT cells at the DE, GT, and PP stages. Three distinct patterns of FOXA2 occupancy were observed. Cluster I regions (149) were bound by FOXA2 from DE to PP stage, cluster II regions (237) were FOXA2-bound at GT and PP stages, while the most predominant group, cluster III (1,888), displayed de novo FOXA2 occupancy in pancreatic progenitors (**Fig. 4e**). We then analyzed the annotations of nearby genes with Genomic Regions Enrichment of Annotations Tool (GREAT) and found that cluster III regions were enriched for terms of neuron and endocrine pancreas development (**Fig. 4f**). Further examination of ATAC-seq, H3K27ac, and 5mC signals in three FOXA2-decreased clusters revealed high levels of DNA methylation accompanied by low levels of enhancer activity and chromatin accessibility in cluster III regions (**Supplementary Fig. 4c**). Taken together, our data imply that a subset of FOXA2 transiently binds to differentiation stage-specific genomic loci with low levels of active chromatin modifications at which additional lineage-specific TFs may be required to facilitate FOXA2 binding and subsequent chromatin remodeling.

To identify potential TFs associated with differential FOXA2 binding we determined unique DNA binding motifs within FOXA2-decreased, increased and stable sites. Interestingly, loci lost FOXA2 binding, particularly in cluster III, were mostly enriched for basic-helix-loop-helix (bHLH) motifs such as the pancreatic endocrine cell fate determinant NEUROD1⁸ (**Fig. 4g, Supplementary Fig. 4d**). In

contrast, regions gained FOXA2 were mostly enriched with motifs of TEAD and GATA family members, and FOXA2-stable sites most prominently feature forkhead and CTCF motifs (**Fig. 4g**). Notably, expression of the bHLH TFs including NEUROD1, PTF1A, and ASCL1 failed to be induced in TET-knockout cells (**Supplementary Fig. 4e**) which likely results in a decrease of FOXA2 binding at genomic loci primarily associated with these TFs. In summary, these data suggest that recruitment of FOXA2 to differentiation stage-specific sites is genetically and epigenetically primed with the cooperation of additional lineage-specific TFs.

3) ChIP-seq for TET1 resulted in only partial overlap with FOXA2 binding sites. It would be important to show quality controls for the specificity of the TET1 antibody.

Response: Because the antibody (Sigma 09-872) we used for TET1-ChIP seq was discontinued, we were not able to perform additional experiment to verify its specificity. We tried to use several other commercially available TET1 antibodies to perform ChIP-qPCR. However, we found similar amplification signals between WT and TET1-knockout cells at TET1 binding sites identified from previous study ². To ensure the best possible accuracy for TET1 occupancy, we decided to not include the TET1 ChIP-seq results in the revised manuscript.

Minor points:

4) lines 165-166 “suggesting that inhibition of TET did not alter the expression levels of pioneer TFs”. Only data for FOXA and GATA are shown.

Response: We appreciate Reviewer 2’s insightful comment. FOXAs and GATAs are the known pioneer factors critical for pancreas development. They begin to express at the endoderm stage and remain high expression levels through differentiation. As suggested, we examined other pioneer TFs, such as PBX1 and TLEs, and found that their expression levels were similar between WT and TKO cells (**Figure 4; for reviewer only**).

Figure 4. Normalized gene expression of *TLE1*, *TLE3*, and *PBX1* in WT and TKO cells at the ES, DE, and PP stages by RNA-seq.

5) lanes 181-182: “consistent with a suggested connection between hypomethylation and activation of transposable elements”. Are transposable elements upregulated in TET KO cells?

Response: We thank the reviewer for pointing this out. Because the sequencing coverage for our RNA-seq was not sufficient to detect the accurate expression of transposable elements, we have revised the statement to now read (line 145-148):

“We observed enrichment of DMCs primarily at intergenic regions and introns (Supplementary Fig. 2d), whereas substantial enrichment of hypo-DMCs was also found in repeat elements, such as long interspersed elements.”

6) lines 192-193: “we analyzed differentially methylated regions (DMRs) by connecting at least three consecutive DMCs”. What is the rationale for this approach?

Response: There are two steps to identify DMRs. We, first, identified differential individual DNA methylation sites (DMCs) among groups, and then defined one DMR that contained at least 3 consecutive DMCs with maximum 300 bp between two consecutive DMCs. The rationale to define DMR is to obtain a greater statistical power if considering adjacent DMCs together as a region than a single CpG site⁹. Similar to our analysis, other software, such as BSmooth, and studies used the same criterion to define DMRs^{10,11}, suggesting our approach will provide enough statistical power to detect biologically meaningful DMRs. To clarify the rationale of identifying DMRs, we have revised the main text (line 158-160) and method sections (line 651-653).

7) Fig 3d,e: This density plot should be accompanied by heat maps showing the signal distribution across the considered genomic regions.

Response: We have added the corresponding heat maps showing signal distribution across the considered genomic regions shown in **Fig. 2d, e** and **Supplementary Fig. 2g** of the revised manuscript.

8) line 200: “demethylation is primarily associated with pioneer TF binding”. Did the author check for the presence of non-pioneering TFs? Here, it would be nice to decipher co-binding of TFs (related to point (2)).

Response: We thank the reviewer for this comment and fully agree that the association of differentiation-specific demethylation with the occupancy of pioneer TFs versus non-pioneer TFs is an interesting point. As suggested by the reviewer, we determined enrichment of PDX1, SOX9, and HNF6 at hyper-DMRs, respectively, by integrating their ChIP-seq data generated from hESC-derived pancreatic progenitors¹² (PDX1, SOX9 and HNF6 are non-pioneer TFs robustly expressed at the pancreatic progenitor stage). The new analysis of non-pioneer TFs was shown in **Fig. 2d** of the revised manuscript. We found that lineage-specific TFs PDX1, SOX9, and HNF6 were also enriched at differentiation-specific hyper-DMRs but to a lesser extent than pioneer TFs FOXA2, GATA4, and GATA6. Taken together, we found a significant portion of hyper-DMRs was distributed in a differentiation-specific

manner, in which they were enriched for both pioneer and lineage-specific TFs and showed remarkable decreases in chromatin activity upon TET inactivation (**Fig 2e**). Given that pioneer TFs are core components of the transcriptional complexes at cis-regulatory elements during differentiation, our data suggest that TETs are essential for enhancers activation and subsequent incorporation of lineage-specific TFs.

A similar question regarding specific interaction of TFs with TET1 was brought up by reviewer #1 (major comment 1), and we thank both reviewers for raising this important issue. To determine potential interaction between TET1 and TFs, we performed co-immunoprecipitation experiment. Due to the lack of proper TET1 antibody for IP, we applied an anti-FLAG antibody to pull down TET1 in the TKO-TET1FL cells which expressed a FLAG-tagged full-length TET1 gene. We have demonstrated that transduction of full-length *TET1* in TET-deficient cells effectively rescues β -cell differentiation (**Fig. 6; Supplementary Fig. 6**). It is therefore reasonable, although not ideal, to apply a FLAG antibody to pull down TET1 and examine its interaction partners in TKO-TET1FL cells. Of particular interest was the observation that only FOXA2 but not FOXA1, GATA6, and SOX17 (non-pioneer TF) was co-precipitated with TET1 (**Fig. 6c, Supplementary Fig. 6d**) suggesting TET1 specifically interacts with FOXA2 in endodermal lineage intermediates.

9) Supplementary Fig 3h and Fig 6a: The signal scale for FOXA2 track is set to 0.87 and 0.59 respectively while in Fig 3g is set to 88. In Fig 3g the Foxa2 signal appears like a robust peak, but in the other figures, the Foxa2 track basically shows background signal (which would become obvious with using the same scale). The claim in lines 874-876 that Fig. 6a represents a FOXA2/TET1 co-bound region needs to be reconsidered.

Response: We thank reviewer for pointing out these issues. We replaced the previously generated FOXA2 ChIP-seq data (GSE117136) ¹ with our own FOXA2 ChIP-seq data. All highlighted regions of FOXA2 binding in genome-browser views of different loci were identified as peaks by MACS2 (**Fig. 2g; Fig. 6d; Supplementary Fig. 2h**).

10) Supplementary Fig 4a: one can hardly appreciate differences between these pictures. A quantification of the signals could be provided.

Response: As suggested, we performed FACS analysis and quantification of the percentage of PDX1⁺ and NKX6.1⁺ in each TET-knockout lines. Almost all mutant lines showed an ability to induce expression of PDX1 and NKX6.1 to the levels comparable to wildtype (**Supplementary Fig. 5a, b**).

11) Fig 4f: how many mice have been used for this experiment? Which statistics? Number of replicates, statistics and p-values should be provided in each figure legend.

Response: We are grateful for Reviewer 2's suggestions and apologize for omissions of the statistics information. We have added the number of mice and statistic information in the figure legend of **Fig. 5d**. Glucose-stimulated human C-peptide secretion experiments were repeated three times with minimal 6 mice in each group.

12) lines 382-83: "FOXA2 can recruit nucleosome remodeling complexes SWI/SNF to alter the surrounding chromatin structure (ref.45)". This reference is wrong and needs to be corrected.

Response: We apologize for the mistake. We have carefully edited and proofread the revised manuscript to ensure that references were correctly added.

13) ChIP-seq protocol described in the methods would need more details.

Response: As requested, we included a detail ChIP-seq protocol in the method sections.

14) lines 594-596: "Integration of analyses. RNA-seq, WGBS, CMS-IP-seq, ATAC-seq and ChIP-seq library preparations were performed as previously described. Detail experimental procedures can be found in Supplementary Materials". The reference is lacking and I could not find additional details of the experimental procedures in the supplements.

Response: We apologize for the mistake. We have carefully edited and proofread the revised manuscript to ensure that all information were correctly added.

Reference

1. Lee, K. *et al.* FOXA2 Is Required for Enhancer Priming during Pancreatic Differentiation. *Cell Rep.* **28**, 382–393 (2019).
2. Verma, N. *et al.* TET proteins safeguard bivalent promoters from de novo methylation in human embryonic stem cells. *Nat. Genet.* **50**, 83–95 (2018).
3. Cirillo, L. A. *et al.* Opening of compacted chromatin by early developmental transcription factors HNF3 (FoxA) and GATA-4. *Mol. Cell* **9**, 279–89 (2002).
4. Rezania, A. *et al.* Reversal of diabetes with insulin-producing cells derived in vitro from human pluripotent stem cells. *Nat. Biotechnol.* **32**, 1121–1133 (2014).
5. Zmuda, E. J., Powell, C. A. & Hai, T. A method for murine islet isolation and subcapsular kidney transplantation. *J. Vis. Exp.* **50**, e2096 (2011).
6. Cong, L. *et al.* Multiplex Genome Engineering Using CRISPR/Cas Systems. *Science* **339**, 819–824 (2013).
7. Cernilogar, F. M. *et al.* Pre-marked chromatin and transcription factor co-binding shape the pioneering activity of Foxa2. *Nucleic Acids Res.* **47**, 9069–9086 (2019).
8. Naya, F. J. *et al.* Diabetes, defective pancreatic morphogenesis, and abnormal enteroendocrine

- differentiation in BETA2/NeuroD-deficient mice. *Genes Dev.* **11**, 2323–2334 (1997).
9. Gu, H. *et al.* Genome-scale DNA methylation mapping of clinical samples at single-nucleotide resolution. *Nat. Methods* **7**, 133–136 (2010).
 10. Hansen, K. D., Langmead, B. & Irizarry, R. A. BSmooth: from whole genome bisulfite sequencing reads to differentially methylated regions. *Genome Biol.* **13**, (2012).
 11. Kottakis, F. *et al.* LKB1 loss links serine metabolism to DNA methylation and tumorigenesis. *Nature* **539**, 390–395 (2016).
 12. Geusz, R. J. *et al.* Pancreatic progenitor epigenome maps prioritize type 2 diabetes risk genes with roles in development. *Elife* **10**, e59067 (2021).

REVIEWER COMMENTS

Reviewer #1 (Remarks to the Author):

The authors addressed all major comments and the manuscript is significantly improved. The work provides novel insights into dynamic DNA methylation and phenotypic consequences during pancreas development. It's a comprehensive and carefully done study with important mechanistic conclusions.

I just have a few minor comments:

- Typo in panel 3d (pancrease should be pancreas)
- Typo in figure legend for 6d (chromatin associability should be accessibility)
- For the clustering of hypo-DHMRs (panel 3a), I don't understand the nomenclature for GT(h)-to-PP vs GT-to-PP(h) and can't find an explanation for it anywhere

Reviewer #2 (Remarks to the Author):

The authors have extensively revised and improved the manuscript. However, with the current set of data it is still possible to argue that TETs are simply necessary for enhancers activation rather than interacting specifically with FOXA and other pioneer TFs as proposed by the authors. The authors identify strongly changed PP development with alterations in TF networks and enhancer usage. In contrast, DE cells seem to display minor changes (at least transcriptionally), although Foxa2 plays important roles in DE differentiation. PP cells appear to be an aberrant endpoint resulting from Tet depletion, but the onset of this phenotype and the specific role of the Foxa2-Tet interaction is largely unclear. Since this is a key point of the manuscript it should be clarified better before publication.

Major comments:

- 1) Specific interaction of Tet1 with Foxa2 needs to be shown more convincingly (Fig6c). It is unclear if this interaction is direct and specific towards Foxa2. Tets are likely to interact with multiple proteins, so it would be expected to see only a small fraction of target TF to interact with Tet (small band for Foxa2 in the Tet-IP), so the signal for the other TFs might be even smaller and below detection limit. Reverse IPs, from the TFs are needed to explore the specificity better. In addition, Flag-Tet1 signal in input should be shown. A Nuclease (like benzonase) treatment should be included to support the physical interaction. Positive controls (proteins known to associate with Tets) would be informative.
- 2) Tet1 ChIP: notoriously TETs proteins are difficult to ChIP with antibodies against the endogenous proteins. Others have used FLAG-TETs (<https://doi.org/10.1186/s13059-018-1464-7>) to circumvent this problem. The authors could use their TET1-flag cell line for TET1 ChIP. The co-localization of Tet1 with Foxa2 binding sites and not other enhancers would be needed to ensure specificity of the Foxa2-Tet axis.
- 3) Presence of lineage specific TFs PDX1, Sox9 and HNF6 at differentiation specific hyper-DMRs (Fig. 2d): the heat maps do not show a sharp and convincing enrichment as for the pioneer TFs and density plots argue for quite some background.

Minor:

- 4) Almost all mutant lines were able to induce the expression of PDX1 and NKX6.1 to levels comparable to WT (lines 290-291, Supplementary Fig. 5a, b): The plot for NKX6.1 does not support this statement
- 5) Avoid the use of words as "dramatically" when describing the data. Examples: line 229; line 245; line 339.

Reviewer #1 (Remarks to the Author):

The authors addressed all major comments and the manuscript is significantly improved. The work provides novel insights into dynamic DNA methylation and phenotypic consequences during pancreas development. It's a comprehensive and carefully done study with important mechanistic conclusions.

I just have a few minor comments:

- Typo in panel 3d (pancrease should be pancreas)
- Typo in figure legend for 6d (chromatin associability should be accessibility)
- For the clustering of hypo-DHMRs (panel 3a), I don't understand the nomenclature for GT(h)-to-PP vs GT-to-PP(h) and can't find an explanation for it anywhere

Response to Reviewer #1

Minor comments:

1. Typo in panel 3d (pancrease should be pancreas)

Response: We apologize for the typo. We have corrected it in **Fig. 3d**.

2. Typo in figure legend for 6d (chromatin associability should be accessibility)

Response: We apologize for the typo. We have corrected it in the main text (line 1274).

3. For the clustering of hypo-DHMRs (panel 3a), I don't understand the nomenclature for GT(h)-to-PP vs GT-to-PP(h) and can't find an explanation for it anywhere

Response: "GT-to-PP(h)" refers to the cluster in which 5hmC levels increased at the GT stage and continuously elevated at the PP stage (Fig. 3c, blue), whereas "GT(h)-to-PP" refers to the cluster in which 5hmC peaked at the GT stage and subsequently decreased at the PP stage (Fig. 3c, brown). We thank the reviewer for this comment and have clarified it in the main text (lines 224-228).

Reviewer #2 (Remarks to the Author):

The authors have extensively revised and improved the manuscript. However, with the current set of data it is still possible to argue that TETs are simply necessary for enhancers activation rather than interacting specifically with FOXA and other pioneer TFs as proposed by the authors. The authors identify strongly changed PP development with alterations in TF networks and enhancer usage. In contrast, DE cells seem to display minor changes (at least transcriptionally), although Foxa2 plays important roles in DE differentiation. PP cells appear to be an aberrant endpoint resulting from Tet depletion, but the onset of this phenotype and the specific role of the Foxa2-Tet interaction is largely unclear. Since this is a key point of the manuscript it should be clarified better before publication.

Major comments:

- 1) Specific interaction of Tet1 with Foxa2 needs to be shown more convincingly (Fig6c). It is unclear

if this interaction is direct and specific towards Foxa2. Tets are likely to interact with multiple proteins, so it would be expected to see only a small fraction of target TF to interact with Tet (small band for Foxa2 in the Tet-IP), so the signal for the other TFs might be even smaller and below detection limit. Reverse IPs, from the TFs are needed to explore the specificity better. In addition, Flag-Tet1 signal in input should be shown. A Nuclease (like benzonase) treatment should be included to support the physical interaction. Positive controls (proteins known to associate with Tets) would be informative.

2) Tet1 ChIP: notoriously TETs proteins are difficult to ChIP with antibodies against the endogenous proteins. Others have used FLAG-TETs (<https://doi.org/10.1186/s13059-018-1464-7>) to circumvent this problem. The authors could use their TET1-flag cell line for TET1 ChIP. The co-localization of Tet1 with Foxa2 binding sites and not other enhancers would be needed to ensure specificity of the Foxa2-Tet axis.

3) Presence of lineage specific TFs PDX1, Sox9 and HNF6 at differentiation specific hyper-DMRs (Fig. 2d): the heat maps do not show a sharp and convincing enrichment as for the pioneer TFs and density plots argue for quite some background.

Minor:

4) Almost all mutant lines were able to induce the expression of PDX1 and NKX6.1 to levels comparable to WT (lines 290-291, Supplementary Fig. 5a, b): The plot for NKX6.1 does not support this statement

5) Avoid the use of words as “dramatically” when describing the data. Examples: line 229; line 245; line 339.

Response to Reviewer #2

Major points:

1) Specific interaction of Tet1 with Foxa2 needs to be shown more convincingly (Fig6c). It is unclear if this interaction is direct and specific towards Foxa2. Tets are likely to interact with multiple proteins, so it would be expected to see only a small fraction of target TF to interact with Tet (small band for Foxa2 in the Tet-IP), so the signal for the other TFs might be even smaller and below detection limit. Reverse IPs, from the TFs are needed to explore the specificity better. In addition, Flag-Tet1 signal in input should be shown. A Nuclease (like benzonase) treatment should be included to support the physical interaction. Positive controls (proteins known to associate with Tets) would be informative.

Response: We greatly appreciate the reviewer for suggesting these experiments. We have performed the reverse co-IP experiment using an anti-FOXA2 antibody followed by immunoblotting of TET1 and have shown the new analysis of co-immunoprecipitation in **Fig. 6c** of the revised manuscript (lines 387-393). The results confirmed the successful pull-down of TET1 together with FOXA2. To exclude the possibility of DNA-protein interaction leading to co-immunoprecipitation, we treated the cell extracts

with nuclease and performed co-IP in the presence of nuclease. Our results demonstrated that nuclease treatment did not result in an appreciable decrease of TET1 proteins pulled down by FOXA2, suggesting a physical interaction between FOXA2 and TET1. The TET1-flag signal in input was shown by immunoblotting with an anti-TET1 antibody (**Fig. 6c**).

Because the IP validated antibodies for GATA6 and FOXA1 were on backorder, we were not able to perform reverse co-IPs on GATA6 and FOXA1. We agree with the reviewer that reverse IPs from the other TFs are needed to explore the specificity of TET1 interaction, yet our current study mainly focuses on the circuit of FOXA2 and TET1. We decided not to include the contexts related to the interaction of TET1 and GATA6/FOXA1/SOX17 in the revised manuscript and will conduct a more comprehensive analysis in our following studies.

2) Tet1 ChIP: notoriously TETs proteins are difficult to ChIP with antibodies against the endogenous proteins. Others have used FLAG-TETs (<https://doi.org/10.1186/s13059-018-1464-7>) to circumvent this problem. The authors could use their TET1-flag cell line for TET1 ChIP. The co-localization of Tet1 with Foxa2 binding sites and not other enhancers would be needed to ensure specificity of the Foxa2-Tet axis.

Response: We are grateful for the reviewer's suggestion and apologize that we have not delivered this point clearly. We have previously reported that 5hmC positively correlates with poised and active enhancers in multiple endodermal lineage intermediates ¹. Thus, we postulate that TET1 proteins localize to most enhancers in pancreatic progenitors, while, at the late stage of pancreas induction, FOXA2 favors TET1 deposition to facilitate local chromatin remodeling. The FOXA2-TET1 axis provides a safeguard against broad gene expression and fine-tunes high threshold cell type-specific gene expression in the pancreatic progenitor domain. We have included more analyses and elaborated on this point in the discussion section (lines 477-531).

Consistent with prior reports that Tet-mediated DNA demethylation mainly occurs at many distally located enhancers ², we found that TET deficiency-induced DNA hypermethylation occurred at enhancers independent of FOXA2 (**Figure 1a; to reviewer only**). Although TET-mediated 5hmC enriched at both FOXA2-bound and unbound enhancers, higher levels of 5hmC were found at the FOXA2-bound enhancers in pancreatic progenitors (**Figure 1b; to reviewer only**). We, therefore, speculate that TET1 proteins co-localize to FOXA2-bound and -unbound enhancers at different levels. As suggested, we performed Flag ChIP-qPCR using TKO-TET1FL cells differentiated at the PP stage. We designed various pairs of primers to detect the enrichment of TET1 at FOXA2-bound (FLAG_P1, _P2, _P3, and _P4) and -unbound (FLAG_P5, _P6, _P7, and _P8) enhancers identified in WT_PP cells (**Figure 1c; to reviewer only**). We found that TET1_flag proteins were enriched at slightly higher levels at enhancers displaying FOXA2 deposition (**Figure 1c; to reviewer only**). However, ectopic the TET1_flag proteins were constitutively expressed throughout differentiation in TKO-TET1FL cells in which TET2 and TET3 were not expressed. It is therefore difficult to conclude whether the

preferential enrichment of TET1 at FOXA2-bound enhancers occurs in cells expressed endogenous TET1, TET2, and TET3. To explore the precise distribution of TETs, a more comprehensive experiment using various loci-specific (*TET1/TET2/TET3*) tag knock-in cell lines coupled with ChIP-seq analysis will be conducted in our following studies.

To further elucidate the FOXA2-TET1 axis, we have performed additional analyses of local chromatin landscape at FOXA2-decreased and -increased sites. Analysis of chromatin accessibility, H3K27ac, and H3K4me1 signal intensity revealed more active and open chromatin at FOXA2-increased sites compared with FOXA2-decreased sites in PP (**Supplementary Fig. 4c**). Although DNA hypermethylation was found in both groups upon TET depletion (**Supplementary Fig. 4d**), H3K27ac, H3K4me1, and ATAC signals were significantly lost at the FOXA2-decreased but not -increased sites (**Supplementary Fig. 4c**). Together with our data showing the most predominant FOXA2-decreased regions (cluster III) displayed de novo FOXA2 occupancy in pancreatic progenitors (Fig. 4e), we believe that these results strengthen our conclusions, as they suggest that TET-mediated hypomethylation at de novo FOXA2 binding loci provides an integration hub to fine-tune cell fate-specific chromatin activation after pancreas induction.

Taken together, our analyses revealed that 1) changes in 5hmC mirror the dynamic binding of FOXA2 in cells differentiated from hESCs through defined lineage intermediates toward pancreatic endocrine fate (Fig. 2), 2) upon TET depletion chromatin activity was markedly decreased at hypohydroxymethylated regions associated with de novo FOXA2 recruitment (Fig. 2d; Fig. 4), and 3) FOXA2 physically interacted with TET1 (Fig. 6c). Moreover, we found that de novo FOXA2-binding sites harbor motifs for bHLH TFs, such as NEUROD1 and PTF1A, which fail to be induced in TET-deficient cells, implying a subset of FOXA2 recruitment-associated chromatin activation requires lineage-specific TFs. We, therefore, postulate that FOXA2 favors TET1 deposition to facilitate local chromatin remodeling after pancreatic lineage induction, while de novo recruitment of FOXA2/TET1 provides a safeguard against broad gene expression and fine-tune high threshold cell type-specific gene expression in the pancreatic progenitor domain. Consistent with this possibility, it has been recently shown that secondary recruitment of FOXA1/2 by lineage-specific TFs to organ-specific enhancers is necessary for chromatin activation and helps establish cell type-specific gene expression within the organ progenitor domain³.

Figure 1. **a** Enrichment profile of methylation ratio (5mC/C) at distally located enhancers showing H3K27ac enrichment (*G-test*; $FDR < 0.05$, > 2.5 kb from TSS) with (left) or without (right) FOXA2 deposition in PP for WT (blue) and TKO (green) cells. **b** Enrichment profile of normalized CMS-IP-seq reads at distally located enhancers showing H3K27ac enrichment (*G-test*; $FDR < 0.05$, > 2.5 kb from TSS) with (dark blue) or without (light blue) FOXA2 binding in WT_PP cells. **c** ChIP-qPCR

showing enrichment of FLAG signal over FOXA2-bound (*FLAG_P1*, *FLAG_P2*, *FLAG_P3*, and *FLAG_P4*) or -unbound (*FLAG_P5*, *FLAG_P6*, *FLAG_P7*, and *FLAG_P8*) enhancers in PP for TKO-TET1FL (blue) and TKO (red) cells. Genome-browser view of selected primers (red arrows) at specific enhancers (highlighted in blue) with or without FOXA2 peaks. *Neg_P1*, *Neg_P2*, and *Neg_P3* are primers included as controls.

3) Presence of lineage-specific TFs PDX1, Sox9 and HNF6 at differentiation specific hyper-DMRs (Fig. 2d): the heat maps do not show a sharp and convincing enrichment as for the pioneer TFs and density plots argue for quite some background.

Response: We appreciate the reviewer's constructive comment and agree with the reviewer that the background of lineage-specific TF ChIP-seq is quite high. We integrated the other existing datasets for PDX1 ChIP-seq (GSE117136) ⁴, which was conducted with similar pancreatic progenitor cells, and analyzed the distribution patterns of PDX1. We found that PDX1 signals showed more convincing enrichment at the "differentiation-specific hyper-DMRs" (**Fig. 2d**). Therefore, we replaced the previous analysis of PDX1 GSE149148 with the PDX1 GSE117136. In terms of SOX9 ChIP-seq, due to relatively low expression levels of SOX9 at the pancreatic progenitor stage, no better SOX9 ChIP-seq datasets were found. We decided to not include the SOX9 results and revised the statement to now read (lines 180-184): Interestingly, we found that pioneer TFs FOXA2, GATA4, and GATA6 were substantially enriched at differentiation-specific hyper-DMRs, while less enrichment of pancreatic lineage-specific TFs such as PDX1 (GSE117136) and HNF6 (GSE149148) was observed (**Fig. 2d**), suggesting that TET-dependent demethylation during pancreatic cell fate commitment occurs at loci primarily associated with pioneer TFs.

Minor points:

4) Almost all mutant lines were able to induce the expression of PDX1 and NKX6.1 to levels comparable to WT (lines 290-291, Supplementary Fig. 5a, b): The plot for NKX6.1 does not support this statement.

Response: We thank the reviewer for pointing out these issues. We have revised the statement to now read (lines 336-338): Most of the mutant lines were able to induce the expression of PDX1 and NKX6.1 to levels comparable to WT except for the TET1/3 knockout line which showed less NKX6.1 expression at the PE stage (Supplementary Fig. 5a, b).

5) Avoid the use of words as “dramatically” when describing the data. Examples: line 229; line 245; line 339.

Response: As suggested, we have deleted the words “dramatically” in the main text (line 250; line 268; line 395).

Reference

1. Li, J. *et al.* Decoding the dynamic DNA methylation and hydroxymethylation landscapes in endodermal lineage intermediates during pancreatic differentiation of hESC. *Nucleic Acids Research* **46**, 2883–2900 (2018).
2. Lu, F., Liu, Y., Jiang, L., Yamaguchi, S. & Zhang, Y. Role of Tet proteins in enhancer activity and telomere elongation. *Genes and Development* **28**, 2103–2119 (2014).
3. Geusz, R. J. *et al.* Sequence logic at enhancers governs a dual mechanism of endodermal organ fate induction by FOXA pioneer factors. *Nature Communications* **2021 12:1** **12**, 1–19 (2021).
4. Lee, K. *et al.* FOXA2 Is Required for Enhancer Priming during Pancreatic Differentiation. *Cell Reports* **28**, 382–393 (2019).

REVIEWERS' COMMENTS

Reviewer #2 (Remarks to the Author):

The authors have partially addressed my concerns. I recommend the following changes to the text.

(1) The specificity of the Foxa2-Tet interaction is still unclear in absence of controls with other transcription factors. I would agree with the authors that their data suggest a physical interaction between Foxa2 and Tet1 (lines 392-393), however, I would modulate the statement in the next sentence (lines 393-394) in a way that Foxa2 can contribute to Tet1 recruitment (as maybe other TFs might help here).

(2) It is unfortunate that the authors could not perform a Tet1 ChIP-seq experiment to demonstrate a preferential enrichment of Tet1 on Foxa2 bound sites. I understand that things are complicated by possible compensation with Tet2/3. Although Foxa2 and Tets are clearly important for PP differentiation, I would not argue that mainly Foxa2 would favor Tet1 deposition at the late stage of pancreas induction.

(3) The discussion statement that "FOXA2/TET1 provides a safeguard against broad gene expression, and therefore fine-tunes the activation of a specific lineage program during developmental transition" would require more specific explanation. Although it clearly makes sense that Foxa2 fine-tunes lineage specific gene expression it is rather unclear what is meant with "safeguard against broad gene expression".

Reviewer #2 (Remarks to the Author):

The authors have partially addressed my concerns. I recommend the following changes to the text.

(1) The specificity of the Foxa2-Tet interaction is still unclear in absence of controls with other transcription factors. I would agree with the authors that their data suggest a physical interaction between Foxa2 and Tet1 (lines 392-393), however, I would modulate the statement in the next sentence (lines 393-394) in a way that Foxa2 can contribute to Tet1 recruitment (as maybe other TFs might help here).

(2) It is unfortunate that the authors could not perform a Tet1 ChIP-seq experiment to demonstrate a preferential enrichment of Tet1 on Foxa2 bound sites. I understand that things are complicated by possible compensation with Tet2/3. Although Foxa2 and Tets are clearly important for PP differentiation, I would not argue that mainly Foxa2 would favor Tet1 deposition at the late stage of pancreas induction.

(3) The discussion statement that "FOXA2/TET1 provides a safeguard against broad gene expression, and therefore fine-tunes the activation of a specific lineage program during developmental transition" would require more specific explanation. Although it clearly makes sense that Foxa2 fine-tunes lineage specific gene expression it is rather unclear what is meant with "safeguard against broad gene expression".

Response to Reviewer #2

(1) The specificity of the Foxa2-Tet interaction is still unclear in absence of controls with other transcription factors. I would agree with the authors that their data suggest a physical interaction between Foxa2 and Tet1 (lines 392-393), however, I would modulate the statement in the next sentence (lines 393-394) in a way that Foxa2 can contribute to Tet1 recruitment (as maybe other TFs might help here).

Response: We appreciate the reviewer's constructive comment and have revised the statement to now read (lines 368-370): Although other TFs might act in addition to FOXA2 to promote TET1 deposition, our data illustrate that FOXA2 contributes to TET1 recruitment at FOXA2-binding loci to mediated demethylation during lineage induction.

(2) It is unfortunate that the authors could not perform a Tet1 ChIP-seq experiment to demonstrate a preferential enrichment of Tet1 on Foxa2 bound sites. I understand that things are complicated by possible compensation with Tet2/3. Although Foxa2 and Tets are clearly important for PP differentiation, I would not argue that mainly Foxa2 would favor Tet1 deposition at the late stage of pancreas induction.

Response: We are grateful for the reviewer's suggestion and have revised the statement to now read (lines 455-460): 3) FOXA2 physically interacted with TET1, suggesting that FOXA2 favors TET1 deposition to facilitate local chromatin remodeling at the late stage of pancreas induction, while de novo

recruitment of FOXA2/TET1 fine-tunes gene induction in the pancreatic progenitor domain. Hence, the identification of preferential enrichment of TET1 in a FOXA2-dependent manner will further strengthen the specificity of the TET1-FOXA2 axis during lineage specification.

(3) The discussion statement that "FOXA2/TET1 provides a safeguard against broad gene expression, and therefore fine-tunes the activation of a specific lineage program during developmental transition" would require more specific explanation. Although it clearly makes sense that Foxa2 fine-tunes lineage specific gene expression it is rather unclear what is meant with "safeguard against broad gene expression".

Response: We apologize that we have not delivered this point clearly and have revised the statement to now read (lines 477-480): We postulate that lineage-specific TF-dependent recruitment of FOXA2/TET1 lowers the threshold of cell-type-specific gene expression, and therefore fine-tunes the activation of a specific lineage program during developmental transition.